# THE NTK ADVERSARY: AN APPROACH TO ADVERSARIAL ATTACKS WITHOUT ANY MODEL ACCESS

## ABSTRACT

We initiate the study of direct generation of adversarial examples for trained neural networks via the analytical tools afforded by *Neural Tangent Kernel (NTK)* theory. In particular, these advances in Deep Learning theory allow us to derive expressions for optimal adversarial perturbations, together with the output of the model on them. This leads to a simple, but powerful strategy for attacking models, which, crucially, does not require any *access* to the model under attack, nor *training* of a substitute model. Instead, we solely exploit information that is available prior to initialization, namely model structure and training data. We experimentally verify the efficacy of our approach, first on models that lie close to the theoretical assumptions (large width, proper initialization, etc.) and, further, on more practical scenarios, with those assumptions relaxed. In addition, we show that our perturbations exhibit strong transferability between models. We provide evidence that, in several cases, there is a close match between our methods and the strongest-to-date adversarial attack, projected gradient ascent, in terms of efficacy and transferability.

## 1 INTRODUCTION

Adversarial examples, a term coined by Szegedy et al. (2014), are inputs of a machine learning model that are carefully crafted so that the model is mistaken on them. Their existence, together with the empirically verified susceptibility of many classifiers to them (Szegedy et al., 2014; Carlini & Wagner, 2017), imply an important security headache for modern machine learning. The ultimate pursuit for robust and immune models calls for an extensive understanding of the different versions of adversarial examples that a model may encounter. In most cases, an adversarial perturbation (an instance of small non random noise) is added to training or test data to form an adversarial example. This procedure is called an *adversarial attack*.

There has been a large body of work on designing such attacks. The most successful attacks exploit model information through its gradient with respect to the input. By perturbing the data in directions of the largest change in loss function, they are able to drop the performance of naive classifiers to almost zero (Goodfellow et al., 2015; Moosavi-Dezfooli et al., 2016; Carlini & Wagner, 2017; Kurakin et al., 2017; Madry et al., 2018). While successful, these gradient-derived attacks require access to the parameters of the trained model - which might be unrealistic for practical applications. To relax this requirement, so called *black-box* attacks only *query* the model. Observation of the model outputs allows to estimate the aforementioned gradients either by training substitute models (Papernot et al., 2016; 2017; Liu et al., 2017) or by finite-difference methods (Chen et al., 2017; Ilyas et al., 2018). Nonetheless, to some extent even these strategies require access to the model.

**Our contribution:** In this work, we leverage recent advances in the theoretical understanding of trained neural networks to devise adversarial data perturbations that, surprisingly, require neither access to the model nor training of a substitute model, and do not use any model information whatsoever, apart from the model *architecture*. We thus reveal an interesting phenomenon: the susceptibility of trained neural networks to adversarial examples does not depend on their specific weights; instead these blind spots may be calculated a-priori, prior to initialization, even before the training starts. Although this is already known to some extent from prior work that trains substitute models, we demonstrate that this notion holds from first principles that only require the a-priori description of the model. Our powerful, highly agnostic attack is derived from Neural Tangent Ker-

nel theory (Jacot et al., 2018; Lee et al., 2019; Arora et al., 2019b), which allows to describe the dynamics of the gradient-descent based training procedure of a neural net as a kernel regression, with an architecture- and data-dependent (but parameter-agnostic) kernel (NTK). At a high-level of abstraction, our approach exploits the NTK description of the function the model has learned at the end of training, and proceeds to maximally perturb this function. To illustrate more precisely, let $(\mathbf{x}, y) \in \mathbb{R}^d \times \{-1, +1\}$ be an input sample, $\boldsymbol{\eta} \in \mathbb{R}^d$ a small perturbation, and $f : \mathbb{R}^d \to \mathbb{R}$ denote the model after training has converged. Then, as we show, in the NTK regime:

$$f(\mathbf{x} + \boldsymbol{\eta}) \approx f(\mathbf{x}) + \boldsymbol{\eta}^T \mathbf{z}, \tag{1}$$

for some $\mathbf{z} \in \mathbb{R}^d$ that depends on the training data and the NTK kernel only. Although the above is derived for regression models, we show that this simple expression naturally leads to a family of adversarial attacks on *classifiers*, depending on the choice of constraints[1] imposed on $\boldsymbol{\eta}$. We also cover the general case of multiclass classifiers, which is similar in spirit to Eq. (1). As our experiments on MNIST and CIFAR suggest, we are able to confuse classifiers with a rate that in some cases approaches current benchmarks that require model access.

In summary, in this work we initiate the study of "NTK-derived" adversarial perturbations and make the following **contributions**:

1. We derive expressions for the output of a converged model on adversarial inputs, based on the theory of the Neural Tangent Kernel (Sec. 3).

2. We propose a family of simple attacks that exploit the aforementioned description. These attacks either depend on analytical expressions afforded by the NTK or on finite approximations of the true quantities (*"empirical"* NTK), obtained by *initialized* replicas of the attacked models (Sec. 3).

3. We experimentally verify the efficacy of our approaches on a simple wide single hidden layer model that operates in the NTK regime (Arora et al., 2019a) (Sec. 4.1) and show that it favorably compares to the strongest benchmark (PGD attack), without any access to model weights.

4. Further, we demonstrate that our attack works in practical settings: fully connected and convolutional, undefended, networks, trained in standard ways (cross entropy minimization, stochastic gradient descent, default initialization, etc) are quite susceptible to NTK noise. We show this both in the case where the noise is generated from the NTK of the model itself, even though the assumptions of NTK theory do not necessarily hold (Sec. 4.2), as well as for *transfer attacks* with adversarial perturbations generated by models within the NTK regime (Sec. 4.3).

## 2 BACKGROUND AND RELATED WORK

Our work bridges two as yet disparate areas: generation of adversarial examples for trained neural networks at inference time and the description of the dynamics of neural network training in the infinite width limit, with the Neural Tangent Kernel.

**Adversarial Examples.** Let $f$ be a classifier, $\mathbf{x}$ be an input (e.g. a natural image) and $y$ its true label (e.g. the image class). Then, given that $f$ is an accurate classifier on $\mathbf{x}$, we define $\tilde{\mathbf{x}}$ to be an adversarial example (Szegedy et al., 2014) for $f$ if the following two hold:

1. $d(\mathbf{x}, \tilde{\mathbf{x}})$ is small for some distance function $d$. Common choices in computer vision applications are the $\ell_p$ norms.

2. $f(\tilde{\mathbf{x}}) \neq y$. That is, the perturbed input is being misclassified.

There are many approaches on how to generate such examples and they are often classified into two large areas: white-box and black-box attacks[2]. White-box attacks assume perfect knowledge of the

---

[1]While we focus on $\ell_\infty$-attacks in this work, we remark that the introduced framework is much more general and the adoption of other similarity measures is equivalent to optimizing Eq. (1) (a linear objective) with the desired constraint on $\boldsymbol{\eta}$.

[2]While we use the word "attack" multiple times, it must be emphasized that the existence of adversarial examples and the search for robust models are not security-only concerns. They align very well with concepts, such as interpretability and human-like perception (Tsipras et al., 2019; Ilyas et al., 2019).

model under-attack and they mainly rely on optimization procedures to maximize the "confusion" of the model (Szegedy et al., 2014; Goodfellow et al., 2015; Moosavi-Dezfooli et al., 2016; Kurakin et al., 2017; Carlini & Wagner, 2017; Madry et al., 2018). The most widely adopted method is called Projected Gradient Descent (Kurakin et al., 2017; Madry et al., 2018) and performs multiple gradient steps on the input space to discover adversarial examples. Formally, if $L(f(\mathbf{x}), y)$ denotes a loss function, then, in the context of $\ell_\infty$ adversaries, setting $\mathbf{x}^0 = \mathbf{x}$ the update formula for PGD is:

$$\mathbf{x}^{t+1} = \Pi_{\mathbf{x}+S}\left(\mathbf{x}^t + \alpha \cdot \mathrm{sgn}(\nabla_{\mathbf{x}^t} L(f(\mathbf{x}^t), y))\right), \quad t = 0, \dots, N-1, \qquad (2)$$

where $\Pi_{\mathbf{x}+S}(\cdot)$ projects the input inside the allowed $\ell_\infty$ ball, and $\alpha$ is a gradient step - hyperparameter. $\mathbf{x}^N$ is the final adversarial example. Such attacks are the most powerful to date. Black-box attacks, on the other hand, are not allowed to estimate the aforementioned gradient directly, but rather are restricted to querying the model without accessing the parameters. A lot of these attacks proceed by collecting pairs of inputs and predictions from the model, then train another classifier $\hat{f}$ on them, attack $\hat{f}$ with white-box attacks and finally use those generated adversarial samples to attack $f$ (Papernot et al., 2016; 2017; Liu et al., 2017). The empirical success of such strategies relies on the phenomenon of *transferability* of adversarial examples (Szegedy et al., 2014; Papernot et al., 2016; Tramèr et al., 2017; Liu et al., 2017): it has been observed that adversarial inputs designed for successfully fooling one classifier, manage to also fool others. A related line of research abandons the need for creating substitute models, at the cost of allowing multiple curated queries to the model to infer the loss function (Chen et al., 2017; Ilyas et al., 2018). Finally, hard-label black box attacks have been considered in the literature, in which an attacker observes only the prediction of $f$ (Brendel et al., 2018; Cheng et al., 2019). The adversarial attack we propose does not fall into any of these categories, since it does not query the target model at all. More related to the scope of this work is the so-called "no-box" setting (Chen et al., 2017; Bose et al., 2020; Li et al., 2020), where an adversary is prohibited from querying the model under-attack. However, to the best of our knowledge, practical attacks under this threat model do train substitute neural networks, making them incomparable to our work.

**Neural Tangent Kernel (NTK).** NTK theory plays an increasing role in the quest to theoretically understand deep learning and the elusive concepts of optimization and generalization in overparametrized networks. It connects deep learning to *Gaussian Processes* and *kernels*. The seminal work of Neal (1996) considered the infinite width limit of neural networks with one hidden layer and showed that at initialization the network is a function drawn from a Gaussian Process with an associated kernel that depends on the activation function. Recently, this limit (and non asymptotic versions of it) has been revisited either for deep random networks where only the last layer is being trained (Lee et al. (2018); de G. Matthews et al. (2018); Novak et al. (2019)) or fully trained networks with gradient descent (Jacot et al., 2018; Lee et al., 2019; Arora et al., 2019b). In the latter case, Jacot et al. (2018) identified a particular quantity that governs the dynamics of the training procedure of a neural network with randomly initialized parameters $\boldsymbol{\theta}$ computing an output $f(\mathbf{x}; \boldsymbol{\theta}) : \mathbb{R}^d \to \mathbb{R}$, which they called *Neural Tangent Kernel*.

The NTK is defined as the expectation of the inner product between the gradients of the function with respect to the random initialization of its parameters $\boldsymbol{\theta}$ evaluated on all inputs $\mathbf{x}_i, \mathbf{x}_j$, i.e.,

$$h(\mathbf{x}_i, \mathbf{x}_j) = \mathbb{E}_{\boldsymbol{\theta}} \nabla_{\boldsymbol{\theta}} f(\mathbf{x}_i; \boldsymbol{\theta})^T \nabla_{\boldsymbol{\theta}} f(\mathbf{x}_j; \boldsymbol{\theta}). \qquad (3)$$

While first studied in fully connected networks, it is possible to define neural tangent kernels for *any* sane architecture (Yang (2020)).

In the infinite width limit[3] (and under appropriate initialization), the training dynamics of the network with infinitesimal learning rate can be described by a differential equation with a constant NTK. Moreover, crucially, in the case of squared loss this gives rise to a closed-form description of the training dynamics with the NTK. For ease of presentation we restrict to the case where the expected network output at initialization is zero; then the expected output of the network at convergence of training becomes (Lee et al., 2018):

$$f^\infty(\tilde{\mathcal{X}}) = \mathbf{H}(\tilde{\mathcal{X}}, \mathcal{X})\mathbf{H}(\mathcal{X}, \mathcal{X})^{-1}\mathcal{Y}, \qquad (4)$$

where $\mathcal{X} \in \mathbb{R}^{n \times d}$ is a collection of $n$ $d$-dimensional training data, $\tilde{\mathcal{X}} \in \mathbb{R}^{n' \times d}$ is a test set and $\mathbf{H} : \mathbb{R}^{n_1 \times d} \times \mathbb{R}^{n_2 \times d} \to \mathbb{R}^{n_1 \times n_2}$ is the NTK (for fixed input datasets, $\mathbf{H}$ is a matrix). For some

---

[3]for non asymptotic statements see Arora et al. (2019b).

models, this kernel can be computed in closed form *analytically* by computing the expectation over the initializations. It can also be approximated by an *empirical* kernel via averaging over a few initialized instances (Jacot et al., 2018; Lee et al., 2019).

Finally, there has also been work on the intersection of the two areas, though with very distinct emphasis to ours. Recent work (Bubeck et al. (2021); Bartlett et al. (2021)) studies the existence of adversarial examples in overparametrized *random* neural networks, exploiting some local linearity of the functions computed by such networks. The only prior work that leverages NTK theory to derive perturbations (published almost concurrently with the preparation of our work) is due to (Yuan & Wu, 2021), yet with different focus and without our modification to the standard approach to the optimization problem, that generates the perturbations (see at the end of Sec. 3). That work is concerned with what is coined as *generalization attacks*: the process of altering the training data distribution to prevent models to generalise on clean data. In contrast, our work analyses trained models to analytically derive their weakness to adversarial perturbations at *inference* time. Besides that, our work introduces analytical expressions for the perturbed data and the output of the network on them, something that does not follow from the methods of Yuan & Wu (2021) (even if extended to the problem we consider).

## 3 ADVERSARIAL PERTURBATIONS ON WIDE NEURAL NETWORKS

Here we present the main methods introduced in this work, assuming the asymptotic conditions on the parameters required for Eq. (4) to hold, and showcasing its implications for adversarially perturbed inputs.

### 3.1 SCALAR OUTPUT

First, we treat the simpler case of networks with scalar output, which allows us to illustrate the various scenarios we examine. We assume the training dataset consists of $n$ $d$-dimensional samples, $\mathcal{X} \in \mathbb{R}^{n \times d}$, with labels $\mathcal{Y} \in \{-1, +1\}^n$.

Suppose we would like to evaluate a model described by Eq. (4) on *slightly* perturbed variations of the original *training* data. Then, slightly abusing notation, we set, $\tilde{\mathcal{X}} = \mathcal{X} + \epsilon$, that is $\tilde{\mathbf{x}}_i = \mathbf{x}_i + \boldsymbol{\eta}_i$ for all $\mathbf{x}_i \in \mathcal{X}$ for small, but unknown, perturbations $\boldsymbol{\eta}_i$. By Taylor expanding around $\mathbf{x}_i$ and neglecting second order terms in the perturbation $\boldsymbol{\eta}_i$, we can write the $ij$-th element of $\mathbf{H}(\tilde{\mathcal{X}}, \mathcal{X})$ as follows:

$$h(\tilde{\mathbf{x}}_i, \mathbf{x}_j) = h(\mathbf{x}_i + \boldsymbol{\eta}_i, \mathbf{x}_j) \approx h(\mathbf{x}_i, \mathbf{x}_j) + \nabla_{\mathbf{x}_i} h(\mathbf{x}_i, \mathbf{x}_j)^T \boldsymbol{\eta}_i. \tag{5}$$

For each row $\underbrace{\mathbf{H}_{i,:}(\tilde{\mathcal{X}}, \mathcal{X})}_{\in \mathbb{R}^{1 \times n}}$ we obtain:

$$\mathbf{H}_{i,:}(\tilde{\mathcal{X}}, \mathcal{X})^T = \mathbf{H}_{i,:}(\mathcal{X}, \mathcal{X})^T + \underbrace{\begin{pmatrix} \nabla_{\mathbf{x}_i} h(\mathbf{x}_i, \mathbf{x}_1)^T \\ \nabla_{\mathbf{x}_i} h(\mathbf{x}_i, \mathbf{x}_2)^T \\ \vdots \\ \nabla_{\mathbf{x}_i} h(\mathbf{x}_i, \mathbf{x}_n)^T \end{pmatrix}}_{\mathbf{A}_i \in \mathbb{R}^{n \times d}} \boldsymbol{\eta}_i. \tag{6}$$

Hence, $\mathbf{H}(\tilde{\mathcal{X}}, \mathcal{X})$ can be written as $\mathbf{H}(\mathcal{X}, \mathcal{X}) + \boldsymbol{\Delta}$ for a perturbation matrix $\boldsymbol{\Delta}$, with $i$-th row $\boldsymbol{\Delta}_{i,:} = \boldsymbol{\eta}_i^T \mathbf{A}_i^T$. Substituting into Eq. (4), we get:

$$f(\tilde{\mathcal{X}}) = (\mathbf{H}(\mathcal{X}, \mathcal{X}) + \boldsymbol{\Delta})\mathbf{H}(\mathcal{X}, \mathcal{X})^{-1}\mathcal{Y} = \mathcal{Y} + \boldsymbol{\Delta}\mathbf{H}(\mathcal{X}, \mathcal{X})^{-1}\mathcal{Y}. \tag{7}$$

Thus, the output of the model on $\tilde{\mathbf{x}}_i$ is:

$$\begin{aligned} f(\tilde{\mathbf{x}}_i) &= y_i + \boldsymbol{\Delta}_i \mathbf{H}(\mathcal{X}, \mathcal{X})^{-1}\mathcal{Y} \\ &= y_i + \boldsymbol{\eta}_i^T \mathbf{A}_i^T \mathbf{H}(\mathcal{X}, \mathcal{X})^{-1}\mathcal{Y} =: y_i + \boldsymbol{\eta}_i^T \mathbf{z}_i, \end{aligned} \tag{8}$$

leading to the linear expression advertised in Eq. (1). The adversarial perturbation $\boldsymbol{\eta}_i$ changes the output by $\boldsymbol{\eta}_i^T \mathbf{z_i} = \boldsymbol{\eta}_i^T \mathbf{A}_i^T \mathbf{H}(\mathcal{X}, \mathcal{X})^{-1}\mathcal{Y}$, an expression which allows us to *compute* the adversarial perturbation to maximally change the output within the desired constraints on $\boldsymbol{\eta_i}$. We call such

adversarial perturbations derived from Eq. (8) *NTK-perturbations*. Notice that here we essentially compute a first-order approximation of the function of the model on perturbed data.

**Regression versus classification** Eq. (4) gives an exact description for regression models with LSE ($L_2$-loss), while adversarial examples typically are studied for classification models. As is standard, we view such regression models as classifiers by thresholding the predictions (i.e. taking the sign of the output in the case of binary $\{-1, 1\}$ classification tasks) or by outputting the maximum prediction (in the case of multiclass problems).

Inspecting Eq. (8), maximal "confusion" of the classification model is achieved by aligning $\boldsymbol{\eta}_i$ with $-y_i \mathbf{z_i}$ (directed towards the decision boundary). In case of the commonly used $\ell_\infty$ restriction, i.e. $\|\boldsymbol{\eta}_i\|_\infty \leq \epsilon$, the optimal adversarial perturbation is given by:

$$\boldsymbol{\eta}_i = -\epsilon y_i \cdot \mathrm{sgn}(\mathbf{A}_i^T \mathbf{H}(\mathcal{X}, \mathcal{X})^{-1} \mathcal{Y}). \tag{9}$$

We remark that the above derivation allows us to exactly compute an optimal adversarial attack for very wide models trained with gradient descent under certain asymptotic conditions necessary for Eq. (4), without any knowledge of its weights. The computation requires an expression for the NTK and its gradient with respect to the training data. For models where an *analytical* expression of the NTK is available, only access to the labeled training data is necessary (see Sec. 4.1). In more complicated models or those that deviate from the assumptions for Eq. (4) one can compute an *empirical* kernel by sampling over kernels at initialization over a few instances and obtain the matrices $\mathbf{A_i}$ with autodifferentiation tools.

**Adversarial perturbations of unseen (test) data** Eq. (8) has been derived for perturbations of the *training* data. Consider now the case when we evaluate Eq. (4) on perturbations of unseen *test* data, that is on $\tilde{\mathcal{X}} + \boldsymbol{\epsilon}$. Then, Eq. (7) becomes:

$$f(\tilde{\mathcal{X}} + \boldsymbol{\epsilon}) = (\mathbf{H}(\tilde{\mathcal{X}}, \mathcal{X}) + \boldsymbol{\Delta})\mathbf{H}(\mathcal{X}, \mathcal{X})^{-1}\mathcal{Y} = f(\tilde{\mathcal{X}}) + \boldsymbol{\Delta}\mathbf{H}(\mathcal{X}, \mathcal{X})^{-1}\mathcal{Y}. \tag{10}$$

Again, solely the second term depends on the perturbation, so we proceed by choosing a maximally perturbing direction as before. The only difference lies in the matrix $\boldsymbol{\Delta}$ that now depends on the test set $\tilde{\mathcal{X}}$ (see App. A.1.1). In practice, an adversary can calculate the NTK $\mathbf{H}(\mathcal{X}, \mathcal{X})$ offline (even a small batch of it, as the experiments suggest) and calculate the optimal perturbation on a new test input $\tilde{\mathbf{x}}_i$ by computing the corresponding row of the matrix $\boldsymbol{\Delta}$. We will use NTK-perturbations derived in this fashion by default when perturbing test data, unless otherwise stated.

**No access to training data** In cases when the adversary has no access to training data (inputs and labels), they can make the assumption that the test data comes from the same distribution, which implies that the model would have learned a similar mapping had it been "trained" on test data instead. They then can directly compute the perturbations via the techniques above, provided they have access to the test set labels (this also holds for multiclass classifiers of Sec. 3.2). We will refer to this method as *training-agnostic* test method.

**Absence of labels** We note that all our methods require access to training (or test) labels, but want to stress that they work even in the setting where the adversary has no access to either labels. To evaluate Eq. (4) and derived expressions, the adversary can "estimate" the labels by observing the output of the attacked model and substituting it instead of the required label vector $\mathcal{Y}$. Such oracular access to the trained model is in contrast to the main point of our work and we don't examine this method in our experiments, but want to point out its existence.

## 3.2 MULTIPLE OUTPUTS

We adapt the derivations of Sec. 3.1 to the setting where the output dimension is larger than one in the underlying regression setting, leading to a multiclass classifier. The derivations can be found in App. A.1.2 and lead to the multi-dimensional analogue of the linear Eq. (1) for $f(\mathbf{x}) \in \mathbb{R}^k$, $\mathbf{y} \in \mathbb{R}^k$:

$$f(\mathbf{x}_i + \boldsymbol{\eta}_i) = \mathbf{y}_i + \begin{pmatrix} \boldsymbol{\eta}_i^T \mathbf{z}_1 \\ \boldsymbol{\eta}_i^T \mathbf{z}_2 \\ \vdots \\ \boldsymbol{\eta}_i^T \mathbf{z}_k \end{pmatrix}. \tag{11}$$

Again, the $\mathbf{z} \in \mathbb{R}^d$ can be computed from the NTK and its derivative as well as the training data labels. Exactly analogous considerations as in the binary case allow to adapt this expression to perturbations of the *test* data, or to substitute access to labeled test data instead of training data.

At this point we have a choice of how to adversarially perturb the classifier to achieve the largest effect on the network output. We present the two most obvious methods, with details in App. A.1.2.

*Max-of-$\ell_1$ perturbation:* Similar in spirit to traditional approaches in adversarial attacks (Carlini & Wagner (2017)) we choose $\boldsymbol{\eta_i}$ such as to most efficiently decrease the correct response $r^*$ while maximally increasing one of the false responses $r \neq r^*$. The solution is given by:

$$\boldsymbol{\eta}_i = \epsilon \mathrm{sgn}(\arg \max_{r=1, r\neq r^*}^{k} \|\mathbf{z}_r - \mathbf{z}_{r^*}\|_1). \tag{12}$$

*Sum-of-$\Delta z$ perturbation:* For one-hot vectors $\mathbf{y}_i$ we could, instead, maximize the cross-entropy between the labels and the new outputs, thus choosing to produce a maximally mixed output. If $r^*$ is the correct label, this yields

$$\boldsymbol{\eta}_i = \epsilon \mathrm{sgn}(\sum_{r\neq r^*}^{n} (\mathbf{z}_r - \mathbf{z}_{r^*})). \tag{13}$$

To summarize, we have shown how the NTK based approach allows us to approximate the model output on perturbed data to first order and then analytically design the optimal perturbation. Note how this contrasts with common approaches, which numerically (and often iteratively) optimize a first order approximation of the *loss* with respect to the perturbation (PGD procedure).

## 4 EXPERIMENTS

In this section, we present experimental evidence on the efficacy of our method. First, we test the performance of NTK-perturbed data on a simple wide single-layer model, that is known to perform very close to what the theory predicts. We present results on binary classification tasks extracted from MNIST and CIFAR10. We also test our multiclass attack on the full MNIST dataset and demonstrate what effect hyperparameters such as batch size, magnitude and method of attack have on the final accuracy. We further provide benchmark comparisons with PGD attacks.

Then, we shift our attention to models encountered more frequently in practice. We try our method on fully connected neural networks initialized with standard techniques and trained by minimizing the cross entropy loss with stochastic gradient descent. Although we violate many of the NTK assumptions, we find that our attacks still yield strong results. Finally, we provide evidence on the transferability of our method: noise crafted from one-hidden-layer networks substantially fools networks commonly used in practice, such as LeNet 300-100 and LeNet 5. Unless stated otherwise, we set the $\ell_\infty$-magnitude of the attack $\epsilon$ equal to 0.3 for MNIST and $8/255$ for CIFAR, as is common practice. All experimental details, hyperparameters and infrastructure can be found in App. A.3.1.

In order to benchmark adversarially generated noise, we compare it to random noise of the same magnitude. When we refer to the *random noise baseline*, we mean impulse noise which with equal probability perturbs each pixel of the image by $\pm\epsilon$ (clipped appropriately to remain inside the allowed input domain; such clipping is applied to all implemented attacks). We report the average and the standard deviation of the random baseline across 5 different random seeds.

### 4.1 RANDOM COMBINATION OF FEATURES MODEL

In the first set of experiments, we test our approach on a simple gradient-trained one-hidden-layer model with an easily computed (analytical) expression for its Neural Tangent Kernel and where experimental evidence confirms that for a sufficiently wide hidden layer the model closely follows theoretical predictions (Arora et al., 2019a). This model of width $m$ computes the following function $f : \mathbb{R}^d \rightarrow \mathbb{R}$:

$$f(\mathbf{x}) = \frac{1}{\sqrt{m}} \sum_{r=1}^{m} a_r \max(\mathbf{w}_r^T \mathbf{x}, 0), \tag{14}$$

where $a_r \sim \mathrm{Unif}(\{-1, +1\})$ are *frozen*, and weights $\mathbf{w}_r$ are initialized from an independent Gaussian distribution $\mathcal{N}(\mathbf{0}, \kappa^2 \mathbf{I})$ for appropriately chosen $\kappa$. We train this model in a regression fashion,

by minimizing the $\ell_2$ loss, and turn it into a classifier by assigning +1 to positive outputs and -1 to non-positive ones. We refer to this model as the Random Combination of Features (*RCF*) model.

The neural tangent kernel $\mathbf{H} \in \mathbb{R}^{n \times n}$ for this model is given by (Cho & Saul, 2009; Arora et al., 2019a) (see App. A.2 for its gradient in order to compute the adversarial perturbation):

$$\mathbf{H}_{ij} = h(\mathbf{x}_i, \mathbf{x}_j) = \left(\frac{1}{2} - \frac{\arccos\left(\frac{\mathbf{x}_i^T \mathbf{x}_j}{\|\mathbf{x}_i\|\|\mathbf{x}_j\|}\right)}{2\pi}\right)\mathbf{x}_i^T \mathbf{x}_j, \ \forall i, j \in [n]. \tag{15}$$

This model generalizes to a function with multiple outputs in the obvious way. To view the regression model with $\ell_2$-loss as a $k$-way classifier, we embed the labels $\in [k]$ as $k$-dimensional one-hot vectors and output the prediction with largest value. Based on NTK theory, the kernel in this multi-class case decomposes; with a copy of the scalar kernel corresponding to each of the outputs.

### 4.1.1 BINARY CLASSIFICATION

We evaluate wide models in Eq. (14) on binary classification tasks extracted from MNIST and CIFAR. On MNIST, we train 45 different models, one for every pair of distinct digits. On CIFAR, we use data from the first two classes ("car" vs. "plane"). We train until convergence. All models on MNIST record almost perfect training and testing performance, while train/test accuracy on CIFAR is 99.35%/ 90.3%. See App. A.3.1 for details and hyperparameters and App. A.3.5 for scalability.

**Analytical NTK** First, we calculate the optimal adversarial perturbation from the analytical expression of the NTK (Eq. (15)), and its gradient with respect to the first input (Eq. (23) in App. A.2)). Table 1 shows the performance for perturbation of the *training* and the unseen (*test*) data. The perturbed data are misclassified with a rate that approaches 100% on CIFAR, and 85% on MNIST. We verify that our models are robust on the random baseline of same magnitude.

**Empirical NTK** We now attack the same model with an empirical approximation of the NTK, estimating the analytical expressions in Eqs. (15) and (23) by initializing replicas of the model under attack using Eq. (3) (with different initialization for each batch - to minimize variance due to choice of the random initialization). See Table 1 for the results. Restricted by hardware limitations, we estimate the kernels using mini-batches of 256 for MNIST and 1500 for CIFAR (see App. A.3.3 for details). The results are similar to the analytical case for MNIST, but not on CIFAR; we find that absence of full data is devastating for the adversary (61% vs 0%), given the higher dimension of this dataset (see, also, App. A.3.4 for more details on the effect of batch size on the success of the attack). Note, however, that this mini-batch approach relaxes our initial assumption that the adversary has access to the whole training distribution; even minimal knowledge suffices to yield strong attacks in some cases!

| **Training** Dataset | NTK (analytical) | NTK (empirical) | Random noise |
|---|---|---|---|
| binary CIFAR | 0.03 | 60.94 | $99.27 \pm 0.06$ |
| binary MNIST - all pairs | $14.74 \pm 16.42$ | $21.77 \pm 14.29$ | $88.43 \pm 15.74$ |
| **Test** Dataset | NTK (analytical) | NTK (empirical) | Random noise |
| binary CIFAR10 | 0.45 | 46.05 | $90.24 \pm 0.06$ |
| binary MNIST - all pairs | $15.25 \pm 16.63$ | $17.81 \pm 16.18$ | $88.37 \pm 15.68$ |

Table 1: RCF model accuracy on NTK-perturbed binary MNIST & CIFAR data. We consider attacks that use the *analytical* kernel, an *empirical* approximation and the *random baseline*. The batch size for empirical kernels is 1500 for CIFAR and 256 (500) for training (test) MNIST, while for the analytical method we use full-batch (the entire dataset). MNIST accuracy is averaged over the 45 different classification tasks.

**NTK versus PGD** To compare our results against a strong benchmark, we include comparisons with PGD attacks across a range of magnitudes (details on the PGD setup can be found in App. A.3.1). Recall that PGD is a strictly more knowledgeable attack with access to the model parameters (see App. A.4 for a comparison of the two attacks). In Fig. 1 (left), we compare the performance of the RCF model against NTK and PGD attacks of varying $\ell_\infty$ magnitude $\epsilon$ on the binary CIFAR test data. It testifies to the strength of our NTK attack, showing that it essentially matches the strongest benchmark attack (PGD) even though it is completely model-parameter agnostic.

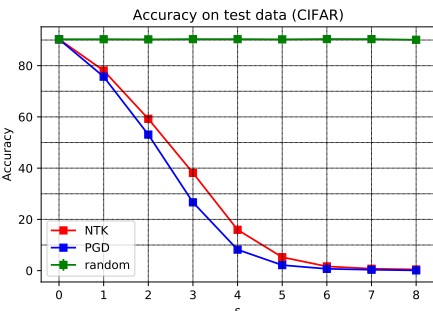 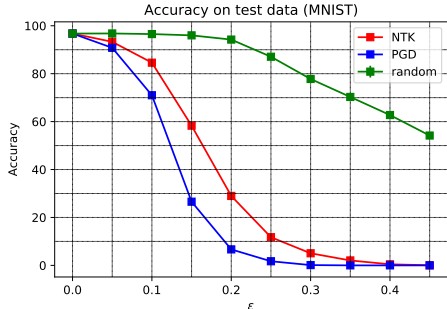

Figure 1: Comparison of our NTK method with PGD attack and random baseline on test data for varying $\ell_\infty$ magnitude of noise $\epsilon$. Left: Binary setting on CIFAR with analytical kernel. $\epsilon$ is given in multiples of $1/255$. Right: Multiclass setting on MNIST with analytical kernel.

### 4.1.2 MULTICLASS CLASSIFICATION

We now train and attack the multiclass version of the RCF model on MNIST. Train and test accuracy for this model are $97.01\%$ and $96.74\%$. See App. A.3.1 for details and hyperparameters. We provide results using the analytical kernel, and refer the interested reader to App. A.3.3 for results with the empirical kernel. While we implement both *max-of-$\ell_1$* and *sum-of-$\Delta z$* perturbations as derived in Sec. 3.2, we adopt *sum-of-$\Delta z$* given that experimental evidence suggests it performs at least as well or better than *max-of-$\ell_1$* (see App. A.3.2 for details). Full kernel experiments (batch size 60K) are presented in Table 2 (see App. A.3.4 and Fig. 2 (right) for the effect of batch size on the success of the attack). We observe that the model is fooled with a rate approaching $95\%$! Moreover, this high success rate only degrades minimally in the case where we do not have access to the training data, but only to labeled test data (*training-agnostic* case, with batch size 10K, see Sec. 3).

| **Training** Dataset | | **Test** Dataset | | |
|---|---|---|---|---|
| NTK | Random noise | NTK | NTK (train-agnostic) | Random noise |
| 4.95 | $77.92 \pm 0.03$ | 5.05 | 8.19 | $77.6 \pm 0.08$ |

Table 2: RCF multiclass attacks with MNIST, using the analytical kernel.

We have also performed the comparison to PGD attacks in the multiclass setting for noise with varying $\ell_\infty$ magnitude $\epsilon$ in Fig. 1 (right), together with the random baseline. As previously observed on binary CIFAR, the proposed attack with the analytical kernels almost matches the performance of PGD, despite the fact that our method never queries the actual model.

### 4.2 FULLY CONNECTED NETWORKS

After a strong validation of our proposed method in the regime where the NTK assumptions hold, we now depart to more practical models, trained with standard techniques (stochastic gradient descent, standard initialization of parameters) directly as classifiers (cross-entropy loss rather than $\ell_2$ regression). To explore how our attack holds up under these relaxed assumptions, we train a family of fully connected networks of varying depth and width on multiclass MNIST. To compute the model-specific NTK perturbation, we approximate the empirical kernel of Eq. (3) by resampling initial parameters and using batches of data. All details can be found in App. A.3.1.

We present results on *training data* in Table 3. Our NTK-attack is surprisingly successful, despite the fact that the models are outside the asymptotic regime where they can be shown to obey Eq. (4), not to mention that even if the dynamics was close to one described by the NTK there is now no guarantee that the NTK estimated with initialized replicas is close to the actual one. These fully converged networks are fooled to below $25\%$ accuracy in almost all models. We observe for deeper models (depth $= 4$ or $6$) that with increasing width the gap to the random baseline widens, matching our expectation since the model is anticipated to operate closer to its tangent kernel with increasing

width (although, as mentioned, there are no theoretical guarantees for this experiment). We leave analytical kernel experiments for these networks (which will allow full batch sizes) for future work.

| width | depth = 2 | | depth = 4 | | depth = 6 | |
|---|---|---|---|---|---|---|
| | NTK | Random | NTK | Random | NTK | Random |
| 100 | 17.91 | $58.35 \pm 0.14$ | 21.91 | $55.87 \pm 0.12$ | 33.36 | $52.91 \pm 0.09$ |
| 1000 | 23.37 | $69.04 \pm 0.24$ | 18.79 | $54.99 \pm 0.30$ | 27.69 | $58.98 \pm 0.08$ |
| 10000 | 25.55 | $71.71 \pm 0.14$ | 19.78 | $64.58 \pm 0.21$ | 22.19 | $60.04 \pm 0.23$ |

Table 3: Accuracy of fully connected nets on NTK-perturbed ($\epsilon = 0.3$) multiclass MNIST *training* data (generated with the *sum-of-$\Delta z$*).

## 4.3 TRANSFERABILITY OF NTK NOISE

Finally, we evaluate how well NTK noise derived from one model "generalizes" into fooling other, well-established models. We train a LeNet 300-100, a LeNet 5 and a wide Residual network on (multiclass) MNIST with standard techniques (see App. A.3.1 for details), and then attack them with NTK-noise generated from the analytical kernel of the RCF model of Sec. 4.1 (see App. A.3.3 for results using the empirical kernel). To compare transferability we also test how well PGD noise coming from the *trained* RCF performs on those architectures, and display accuracy of the random baseline to quantify robustness. Table 4 presents results on training and test data. Surprisingly, we find that adversarial examples crafted from the kernel of the simple RCF model successfully fool models not believed to be captured by NTK theory. We also observe that our attacks carry transferability characteristics comparable to those of the most powerful white-box attack.

| noise (**train**) | RCF | LeNet 300-100 | LeNet 5 | WideResNet |
|---|---|---|---|---|
| RCF | 4.95 | 3.7 | 24.94 | 60.96 |
| PGD | 0.15 | 0.59 | 18.51 | 62.75 |
| Random | $77.92 \pm 0.03$ | $58.48 \pm 0.11$ | $80.29 \pm 0.09$ | $96.04 \pm 0.09$ |
| Clean | 97.01 | 99.96 | 99.82 | 99.78 |
| noise (**test**) | RCF | LeNet 300-100 | LeNet 5 | WideResNet |
| RCF | 5.05 | 3.74 | 25.65 | 64.5 |
| RCF (train-agnostic) | 8.19 | 8.63 | 27.44 | 66.38 |
| PGD | 0.12 | 0.61 | 18.76 | 63.45 |
| Random | $77.6 \pm 0.08$ | $59.38 \pm 0.17$ | $80.52 \pm 0.25$ | $95.87 \pm 0.15$ |
| Clean | 96.74 | 98.02 | 98.62 | 98.29 |

Table 4: Transfer results on MNIST (train & test) with NTK-noise generated from the analytical kernel of the RCF model with the sum-of-$\Delta z$ attack. We compare with PGD perturbed data produced by a trained RCF with cross-entropy loss and we, also, report accuracy on clean data.

## 5 DISCUSSION

In this work, we initiate the study of adversarial examples for trained networks through the lens of Neural Tangent Kernel theory, by deriving analytical expressions for optimal perturbations, and, subsequently, proposing attacks based on them. Our experiments demonstrate the efficacy of the approach in a variety of settings and yield results comparable to the most powerful "white-box" attack (PGD), without model access nor training of a substitute model. All these findings suggest that the methods we introduce provide us with a remarkable, almost off-the-shelf, resource of high-quality adversarial examples.

By connecting NTK and adversarial perturbations, our work demonstrates that it is possible to argue analytically about properties of adversarial examples on deep neural networks. We expect these tools, together with additional insights from NTK theory, to shed some light on the puzzling transferability phenomenon of adversarial examples (by studying the properties of kernels of different architectures) and to facilitate the training of robust neural networks with much less compute (by solving, for example, the robust optimization problem analytically).

## REPRODUCIBILITY

In keeping with conference guidelines, we have tried to ensure that our experimental results were free of any bias that would misconstrue the results. As specified in the paper, all experiments were run over multiple random seeds, on a variety of benchmarks using publicly available software and hardware. We have implemented the methods using auto-differentiation tools from Pytorch autograd. The code for the experiments is self contained and references appropriate instructions to install the relevant software dependencies. A copy of the experimental code will be made publicly available.

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

## A APPENDIX

### A.1 DERIVATION OF ADVERSARIAL PERTURBATIONS

#### A.1.1 PERTURBATIONS OF UNSEEN TEST DATA

Here give the explicit expression of the matrix $\Delta$ in the case of unseen test data:

$$
\boldsymbol{\Delta}_{i,:} = \boldsymbol{\eta}_i^T \begin{pmatrix} \nabla_{\tilde{\mathbf{x}}_i} h(\tilde{\mathbf{x}}_i, \mathbf{x}_1)^T \\ \nabla_{\tilde{\mathbf{x}}_i} h(\tilde{\mathbf{x}}_i, \mathbf{x}_2)^T \\ \vdots \\ \nabla_{\tilde{\mathbf{x}}_i} h(\tilde{\mathbf{x}}_i, \mathbf{x}_n)^T \end{pmatrix}^T
\tag{16}
$$

#### A.1.2 MULTICLASS DERIVATIONS

Here we derive the expressions for Eq. (11).

While we remain with $\mathcal{X} \in \mathbb{R}^{n \times d}$, the other quantities change as $\mathcal{Y} \in \mathbb{R}^{nk}$, $f(\mathcal{X}) \in \mathbb{R}^{nk}$ and $\mathbf{H}(\mathcal{X}, \mathcal{X}) \in \mathbb{R}^{nk \times nk}$, i.e. for each data pair $(\mathbf{x}_i, \mathbf{x}_j)$ we have $\mathbf{H}(\mathbf{x}_i, \mathbf{x}_j) \in \mathbb{R}^{k \times k}$. Let $\mathbf{H}_{lm}(\mathbf{x}_i, \mathbf{x}_j)$

denote the entry of $\mathbf{H}(\mathbf{x}_i, \mathbf{x}_j)$ that corresponds to the $l$-th and the $m$-th output of the model (evaluated at $\mathbf{x}_i$ and $\mathbf{x}_j$). Then, with similar reasoning that led to Eq. (5) we now obtain:

$$\mathbf{H}_{lm}(\mathbf{x}_i + \boldsymbol{\eta}, \mathbf{x}_j) \approx \mathbf{H}_{lm}(\mathbf{x}_i, \mathbf{x}_j) + \nabla_{\mathbf{x}_i} \mathbf{H}_{lm}^T(\mathbf{x}_i, \mathbf{x}_j)\boldsymbol{\eta}. \tag{17}$$

For the prediction of the model on the whole dataset, we have:

$$f(\tilde{\mathbf{X}}) = \mathcal{Y} + \underbrace{\boldsymbol{\Delta}}_{\in \mathbb{R}^{nk \times nk}} (\mathbf{H}(\mathbf{X}, \mathbf{X}))^{-1}\mathcal{Y}, \tag{18}$$

which for a given sample $\mathbf{x}_i$ gives:

$$f(\mathbf{x}_i + \boldsymbol{\eta}_i) = \underbrace{\mathbf{y}_i}_{\in \mathbb{R}^k} + \boldsymbol{\Delta}_{ik:(i+1)k,:} (\mathbf{H}(\mathbf{X}, \mathbf{X}))^{-1} \mathcal{Y}, \tag{19}$$

where $\boldsymbol{\Delta}_{(i-1)k:ik,:}$ is equal to

$$\underbrace{\left( \begin{matrix} \overbrace{(\nabla_{\mathbf{x}_i}\mathbf{H}_{11}(\mathbf{x}_i, \mathbf{x}_1) & \dots & \nabla_{\mathbf{x}_i}\mathbf{H}_{1k}(\mathbf{x}_i, \mathbf{x}_1) & \nabla_{\mathbf{x}_i}\mathbf{H}_{11}(\mathbf{x}_i, \mathbf{x}_2) & \dots & \nabla_{\mathbf{x}_i}\mathbf{H}_{1k}(\mathbf{x}_i, \mathbf{x}_n))^T}^{\in \mathbb{R}^{nk \times d}} \boldsymbol{\eta}_i \\ (\nabla_{\mathbf{x}_i}\mathbf{H}_{21}(\mathbf{x}_i, \mathbf{x}_1) & \dots & \nabla_{\mathbf{x}_i}\mathbf{H}_{2k}(\mathbf{x}_i, \mathbf{x}_1) & \nabla_{\mathbf{x}_i}\mathbf{H}_{21}(\mathbf{x}_i, \mathbf{x}_2) & \dots & \nabla_{\mathbf{x}_i}\mathbf{H}_{2k}(\mathbf{x}_i, \mathbf{x}_n))^T \boldsymbol{\eta}_i \\ & & & \vdots \\ (\nabla_{\mathbf{x}_i}\mathbf{H}_{k1}(\mathbf{x}_i, \mathbf{x}_1) & \dots & \nabla_{\mathbf{x}_i}\mathbf{H}_{kk}(\mathbf{x}_i, \mathbf{x}_1) & \nabla_{\mathbf{x}_i}\mathbf{H}_{k1}(\mathbf{x}_i, \mathbf{x}_2) & \dots & \nabla_{\mathbf{x}_i}\mathbf{H}_{kk}(\mathbf{x}_i, \mathbf{x}_n))^T \boldsymbol{\eta}_i \end{matrix} \right)}_{\in \mathbb{R}^{k \times nk}}. \tag{20}$$

*Derivation of Max-of-$\ell_1$ perturbation:* Similar in spirit to traditional approaches in adversarial attacks (see e.g. Carlini & Wagner (2017)), we solve:

$$\boldsymbol{\eta}_i = \arg \max_{\|\boldsymbol{\eta}_i\|_\infty \leq \epsilon} \max_{r=1, r\neq r^\star}^k f_r(\mathbf{x}_i + \boldsymbol{\eta}_i) - f_{r^\star}(\mathbf{x}_i + \boldsymbol{\eta}_i),$$

where $r^\star = \arg\max \mathbf{y}_i$, i.e. we chose $\boldsymbol{\eta}_i$ such as to most efficiently decrease the correct response while maximally increasing one of the false responses. The solution is given by:

$$\begin{aligned} \boldsymbol{\eta}_i &= \arg \max_{\|\boldsymbol{\eta}_i\|_\infty \leq \epsilon} \max_{r=1, r\neq r^\star}^k \boldsymbol{\eta}_i^T \mathbf{z}_r - \boldsymbol{\eta}_i^T \mathbf{z}_{r^\star} \\ &= \arg \max_{\|\boldsymbol{\eta}_i\|_\infty \leq \epsilon} \max_{r=1, r\neq r^\star}^k \boldsymbol{\eta}_i^T (\mathbf{z}_r - \mathbf{z}_{r^\star}) \\ &= \epsilon \mathrm{sgn}(\arg \max_{r=1, r\neq r^\star}^k \|\mathbf{z}_r - \mathbf{z}_{r^\star}\|_1). \end{aligned} \tag{21}$$

*Derivation of Sum-of-$\Delta z$ perturbation:* For one-hot vectors $\mathbf{y}_i$ we could, instead, maximize the cross-entropy between the labels and the new outputs, thus choosing to produce a maximally mixed output:

$$\begin{aligned} L_{ce}(f(\mathbf{x}_i + \boldsymbol{\eta}), \mathbf{y}_i) &= -\sum_{r=1}^k y_i^{(r)} \log \left( \frac{e^{y_i^{(r)} + \boldsymbol{\eta}_i^T \mathbf{z}_r}}{\sum_{r'=1}^k e^{y_i^{(r')} + \boldsymbol{\eta}_i^T \mathbf{z}'_r}} \right) \\ &= -\log \left( \frac{e^{1 + \boldsymbol{\eta}_i^T \mathbf{z}_{r^\star}}}{\sum_{r=1, r\neq r^\star}^k e^{\boldsymbol{\eta}_i^T \mathbf{z}_r} + e^{1 + \boldsymbol{\eta}_i^T \mathbf{z}_{r^\star}}} \right) \\ &= \log \left( \sum_{r\neq r^\star} e^{\boldsymbol{\eta}_i^T (\mathbf{z}_r - \mathbf{z}_{r^\star}) - 1} + 1 \right) \end{aligned} \tag{22}$$

which amount to maximizing

$$\sum_{r\neq r^\star} e^{\boldsymbol{\eta}_i^T (\mathbf{z}_r - \mathbf{z}_{r^\star})}.$$

For small perturbations we can develop the exponential to first order[4], which leads to finding the maximum of

$$\boldsymbol{\eta}_i^T \sum_{r \neq r^\star} (\mathbf{z}_r - \mathbf{z}_{r^\star})$$

This yields $\boldsymbol{\eta}_i$ equal to $\epsilon \mathrm{sgn}(\sum_{r \neq r^*}^n (\mathbf{z}_r - \mathbf{z}_{r^\star}))$.

## A.2 ANALYTICAL RCF KERNEL

We give the formula for the gradients of the different elements of the NTK for the RCF model considered in the experiments.

$$\nabla_{\mathbf{x}_i} h(\mathbf{x}_i, \mathbf{x}_j) = (\frac{1}{2} - \frac{\arccos\left(\frac{\mathbf{x}_i^T \mathbf{x}_j}{\|\mathbf{x}_i\|\|\mathbf{x}_j\|}\right)}{2\pi})\mathbf{x}_j + \nabla_{\mathbf{x}_i}\left(\frac{1}{2} - \frac{\arccos\left(\frac{\mathbf{x}_i^T \mathbf{x}_j}{\|\mathbf{x}_i\|\|\mathbf{x}_j\|}\right)}{2\pi}\right)\mathbf{x}_i^T \mathbf{x}_j$$

$$= (\frac{1}{2} - \frac{\arccos\left(\frac{\mathbf{x}_i^T \mathbf{x}_j}{\|\mathbf{x}_i\|\|\mathbf{x}_j\|}\right)}{2\pi})\mathbf{x}_j + \frac{1}{2\pi\sqrt{1 - (\frac{\mathbf{x}_i^T \mathbf{x}_j}{\|\mathbf{x}_i\|\|\mathbf{x}_j\|})^2}}\nabla_{\mathbf{x}_i}\left(\frac{\mathbf{x}_i^T \mathbf{x}_j}{\|\mathbf{x}_i\|\|\mathbf{x}_j\|}\right)\mathbf{x}_i^T \mathbf{x}_j$$

$$= (\frac{1}{2} - \frac{\arccos\left(\frac{\mathbf{x}_i^T \mathbf{x}_j}{\|\mathbf{x}_i\|\|\mathbf{x}_j\|}\right)}{2\pi})\mathbf{x}_j + \frac{1}{2\pi\sqrt{1 - (\frac{\mathbf{x}_i^T \mathbf{x}_j}{\|\mathbf{x}_i\|\|\mathbf{x}_j\|})^2}}\frac{\mathbf{x}_j\|\mathbf{x}_i\|\|\mathbf{x}_j\| - \mathbf{x}_i^T \mathbf{x}_j \frac{\mathbf{x}_i\|\mathbf{x}_j\|}{\|\mathbf{x}_i\|}}{\|\mathbf{x}_i\|^2\|\mathbf{x}_j\|^2}\mathbf{x}_i^T \mathbf{x}_j.$$

$$\tag{23}$$

Note that one must pay attention when evaluating the diagonal entries (or any entry whose corresponding vectors are aligned), because the $\arccos$ function is not differentiable at 1. To proceed, one has the option to do an analytic expansion over 1, and treat this case separately. In practice, we truncate all inner products between $-1 + \delta$ and $1 - \delta$ for some small $\delta > 0$ (e.g. $\delta = 1e - 5$), since we found this makes the method numerically stable.

## A.3 EXPERIMENTAL DETAILS

### A.3.1 SETUP

For all models, we used GPU enabled hardware to expedite computational time. Our experiments were performed on a SLURM cluster enabled with NVIDIA V100 Tesla and RTX 8000 GPUs, using the PyTorch package.

In all data sets and all experiments, we drop the last batch if its length is smaller than the others.

**RCF Models**  In all experiments, we set the width $m$ of the models equal to $10000$ and the variance of the random initialization, $\kappa^2$, to $0.01^2$. To train the models, we use (full-batch) gradient descent optimizing the quadratic loss with learning rate equal to 0.01. On MNIST, all models are trained for at least 5000 epochs, and then we stop training when training accuracy crosses $99.8\%$. They record test accuracy between $99.17\%$ and $99.91\%$. On CIFAR, we train the model for 200k epochs. For the multiclass experiment on MNIST, we train a model with the same hyperparameters for 200k epochs. It recorded $97.01\%$ and $96.74\%$ accuracy on train and test data, respectively. The empirical kernels were approximated using solely one replica of the models *per* batch. Given that defining a network is essentially cost-free (a few lines of code without any/much computational cost), we make the decision to draw a new random seed at each batch in order to alleviate the randomness factor.

**Fully Connected Nets**  We initialize the parameters of the linear layers of the FC networks using the default PyTorch method, i.e. a uniform distribution $\mathcal{U}(-\sqrt{k}, \sqrt{k}), k = \frac{1}{\text{in features}}$. The models are trained with stochastic gradient descent using mini-batches of 256, optimizing the cross entropy loss. All models are trained to convergence (see Table 5 for accuracies). We estimate the NTKs

---

[4]The resulting expression for the maximum also holds when developing to second order.

empirically, by defining exact replicas of the models to approximate equation 3. In the smaller nets, we use 3 models initialized with different random seeds for each batch for the approximation, and just one replica per batch on the largest ones (width $= 10000$ and depth equal to 4 or 6). Limited by the heavy computations of the autodifferentiaton tool to estimate the NTK related quantities, we set batch size equal to 64. The $\ell_\infty$-magnitude of the perturbations is set to $\epsilon = 0.3$. Due to limitations in compute we restrict ourselves to produce results on $20\%$ of the training set.

| model | Train | Test |
|---|---|---|
| (2, 100) | 99.02 | 97.69 |
| (2, 1000) | 99.03 | 97.94 |
| (2, 10000) | 99.01 | 97.83 |
| (4, 100) | 99.05 | 97.36 |
| (4, 1000) | 99.06 | 97.67 |
| (4, 10000) | 99.09 | 97.73 |
| (6, 100) | 99.21 | 96.52 |
| (6, 1000) | 99.51 | 97.1 |
| (6, 10000) | 99.34 | 97.42 |

Table 5: Train and test accuracy on fully connected nets (MNIST).

**Transferability**  To generate the NTK-noise we use the sum-of-$\Delta z$ method using mini-batches of size 256 and $\epsilon = 0.3$. All models were trained with stochastic gradient descent on the $\ell_2$ loss with batch size set to 256. Learning rate was set to 0.01. LeNet 300-100 was trained for 500 epochs, recording at the end of training $99.96\%$ train accuracy and $98.02\%$ on test. LeNet 5 was trained for 250 epochs, recording at the end of training $99.82\%$ train and $98.62\%$ test accuracy. ResNet belongs in the family of wide ResNets 50-2, and it was trained for 100 epochs recording at the end of training $99.78\%$ training and $98.29\%$ test accuracy.

**PGD attack details**  PGD attacks as in Eq. (2) optimize the cross-entropy loss for 40 iterations with step size $\alpha = 0.01$ on MNIST and $\alpha = 2/255$ on CIFAR.

### A.3.2  COMPARISON OF DIFFERENT MULTICLASS ATTACKS

Here we compare the *max-of-$\ell_1$* and *sum-of-$\Delta z$* methods for attacking multiple output models as derived in Sec. 3.2, in the setting of the RCF model on MNIST to illustrate our adoption of *sum-of-$\Delta$*. First, we evaluate both methods on test data using the analytical kernel with batch size 64 (i.e. using only the first 64 training images), and we obtain $56.47\%$ accuracy for *max-of-$\ell_1$* and $42.08\%$ for *sum-of-$\Delta z$*. Then, we compare them on training data with empirical kernels with batch size 64, obtaining $37.57\%$ and $28.37\%$[5], respectively. *Sum-of-$\Delta z$* is clearly a better choice for both types of kernel. We performed the comparison in the small batch regime, since this the case of interest in Sec. 4.2 where empirical kernels are estimated using little data. For larger batch sizes, we found that the two methods perform similarly (see Fig. 2 (right) in App. A.3.4).

### A.3.3  EMPIRICAL KERNEL RESULTS OF SEC. 4.1.2 AND SEC. 4.3

Here we provide the results for the experiments on the multitask classification task of MNIST using empirical kernels to compute the adversarial perturbations.

First, it must be noted that we attack the model in *batches*. If, for example, the batch size is set to 256 then we first calculate the optimal perturbations for the first 256 images using only these images for the kernel expressions $\mathbf{H}(\mathcal{X}, \mathcal{X})$ and $\mathbf{A}_i$ in Eq. (9), then repeat for the next 256 and so on. In the case of training data, we progressively use all the training dataset, but in the case of test set attacks we *solely* use the first 256 images of the training dataset. This is equivalent to assuming that the model was only trained on this batch of data.

---

[5]A note to the careful reader: these results are better than the corresponding ones with the analytical kernel, although that is not the case elsewhere (analytical results are always superior). This is attributed to the different approach we take regarding the batch size in train vs test data, which is important with small batch sizes. See the second paragraph of A.3.3

In Table 6, we present results for batch size 256 (column 3) and compare to the random baseline (column 4) on training data. These two results are then presented for the test set (columns 5 and 6).

| sum $\Delta z$ (256) | random train | sum $\Delta z$ test (256) | random test |
|---|---|---|---|
| 24.31 | $77.92 \pm 0.03$ | 24.81 | $77.6 \pm 0.08$ |

Table 6: RCF multiclass empirical attacks with MNIST.

Table 7 contains the results on the transferability experiment of Sec. 4.3 using an empirical kernel estimated with batch size set to 256. The results are inferior to those of Table 4, which is due to the small batch size (see Sec. A.3.4 for an ablation study).

| noise (**train**) | RCF | LeNet 300-100 | LeNet 5 | WideResNet |
|---|---|---|---|---|
| RCF | 24.31 | 14.65 | 39.83 | 74.47 |

| noise (**test**) | RCF | LeNet 300-100 | LeNet 5 | WideResNet |
|---|---|---|---|---|
| RCF (train-agnostic) | 24.81 | 14.74 | 40.54 | 74.04 |
| RCF | 32.25 | 19.7 | 48.85 | 77.38 |

Table 7: Transfer results on MNIST (train & test) with NTK-noise (empirical kernel with batch size 256) generated from the RCF model with the sum-of-$\Delta z$ attack.

### A.3.4 BATCH SIZE EFFECT

Section 4 contains experiments that employ both empirical and analytical kernels. However, current computational limitations do not allow a fair comparison between the two approaches, as the empirical kernel is being calculated from relatively small batch sizes comprising only a subset of the entire dataset. In order to study the effect of batch size on the quality of adversarial examples, we hence turn here to the *analytical* kernel with a variety of batch sizes.

First, Table 8 presents the trendline for the binary task of CIFAR with the RCF model considered in Sec. 4.1.1, using test data. We observe a significant effect of batch size on the success of the attack. For batch size 1500, we see accuracy of $39.05\%$ (which is close to the $46.05\%$ obtained from the empirical kernel using the same batch size, Table 1), but as the batch size increases, the success rate of the NTK attack quickly approaches $100\%$.

| batch size | Accuracy |
|---|---|
| 1500 | 39.05 |
| 4000 | 5.95 |
| 7000 | 1.05 |
| 10000 | 0.45 |

Table 8: Test accuracy vs batch size. Binary CIFAR with analytical kernel of RCF.

The above study was performed with $\epsilon = 8/255$. To better understand how the batch size affects the success of our method for a variety of $\epsilon$, in Fig. 2 we reproduce Fig. 1 enriched with curves that correspond to different batch sizes. In Fig. 2 (right) solid lines correspond to *sum-of-$\Delta z$*, dashed lines to *max-of-$\ell_1$*, and all attacks use analytical kernels (with the exception of the black line). Notice that in the small batch regimes the *sum-of-$\Delta z$* is clearly a better strategy for crafting adversarial examples (which is why we adopt it in the experiments of Sec. 4.2 and Sec. 4.3, see also App. A.3.2), but, given more data, the two approaches seem to perform equally well.

### A.3.5 SCALABILITY

Our experiments with analytical kernels, which are computable for a wide variety of models, are fast, since they require only a few tensor multiplications, even for very large batch sizes (though with growing memory requirements). This is, in fact, an additional advantage of our NTK approach.

Most of our experiments with empirical kernels where run as proofs-of-principle on our own non-optimized auto-differentiation tools of PyTorch. The resulting framework is computationally slow

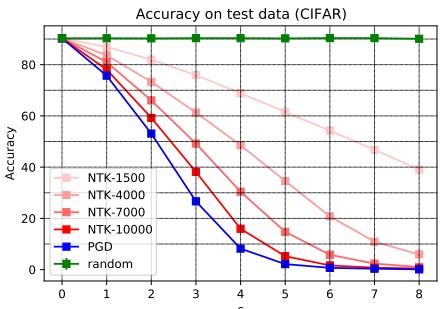 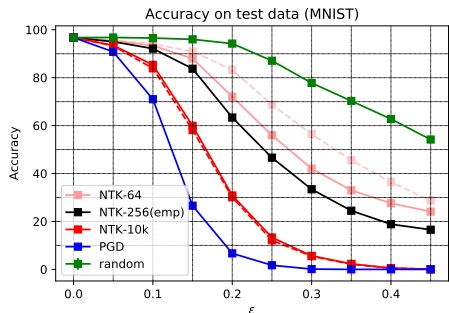

Figure 2: Role of batch size on the test accuracy for varying $\ell_\infty$ magnitude of noise $\epsilon$. Left: Binary setting on CIFAR with analytical kernel. $\epsilon$ is given in multiples of $1/255$. Right: Multiclass setting on MNIST with analytical/empirical kernel. Solid lines correspond to *sum-of-*$\Delta z$, dashed lines to *max-of-*$\ell_1$.

(even when GPUs are employed), hence not allowing for full experimentation with very large batch sizes. However, going forward, utilization of the JAX library (Bradbury et al., 2018) together with the recent Neural Tangents one (Novak et al., 2020) (as employed for example in the work of Yuan & Wu (2021)) will allow the assessment of our attacks with empirical kernels on larger models and datasets, thus opening our methods to any architecture without the need for deriving the analytical expressions.

### A.4 Why PGD perturbations use strictly more knowledge than NTK perturbations

Comparing PGD attacks with our methods and arguing that the former require more knowledge might be confusing, since the assumed threat models are different. Indeed, PGD attacking a network necessitates access to the model and its weights, while in our methods we assume access to the training data (and some information about the model architecture), but not to the model. Hence, the two threat models might seem incomparable. Here, we argue why, for the purposes of this work, this is not the case.

Current research on adversarial machine learning focuses on a security paradigm, where a threat model is assumed, an attack is proposed, and subsequently a defense is created to mitigate the problem. In order to try to understand the origin of the existence of adversarial examples, it may be useful to abandon the distinction between an attacker and a defender and view their generation as a one way process. Viewed through this lens, for a PGD "attack" (or any "white-box attack"), we train a model, the model learns from the data, and then by calculating the gradients of the model, we craft adversarial examples. Notice that this process uses all the information about the data that the model has stored in its weights: it **does** use the data. One contribution of our work is the observation that the final weights of the model do not contribute to the generation of these examples (at least for the experiments conducted and those models in the NTK regime). The above elaborates our claim in the main text that a "white-box attack" is a strictly more knowledgeable procedure than the ones introduced in this work.

### A.5 Adversarial Images

Here we provide some sample images from CIFAR and MNIST, perturbed with our methods. The perturbation is computed from the analytical kernel of the RCF model.

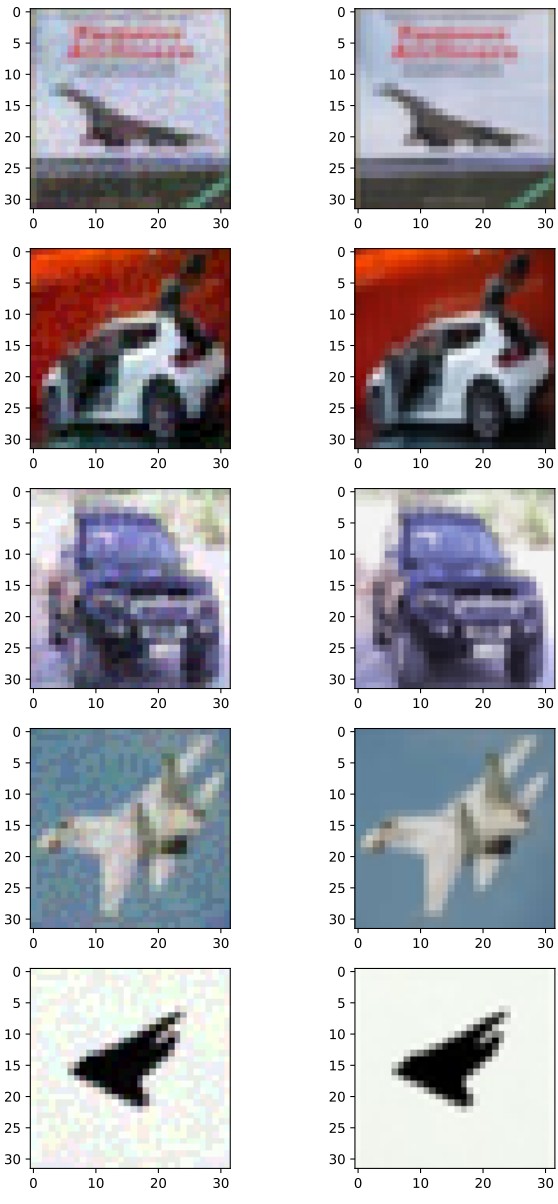

Figure 3: CIFAR images. Perturbed images (left) and clean counterparts (right). $\epsilon = 8/255$

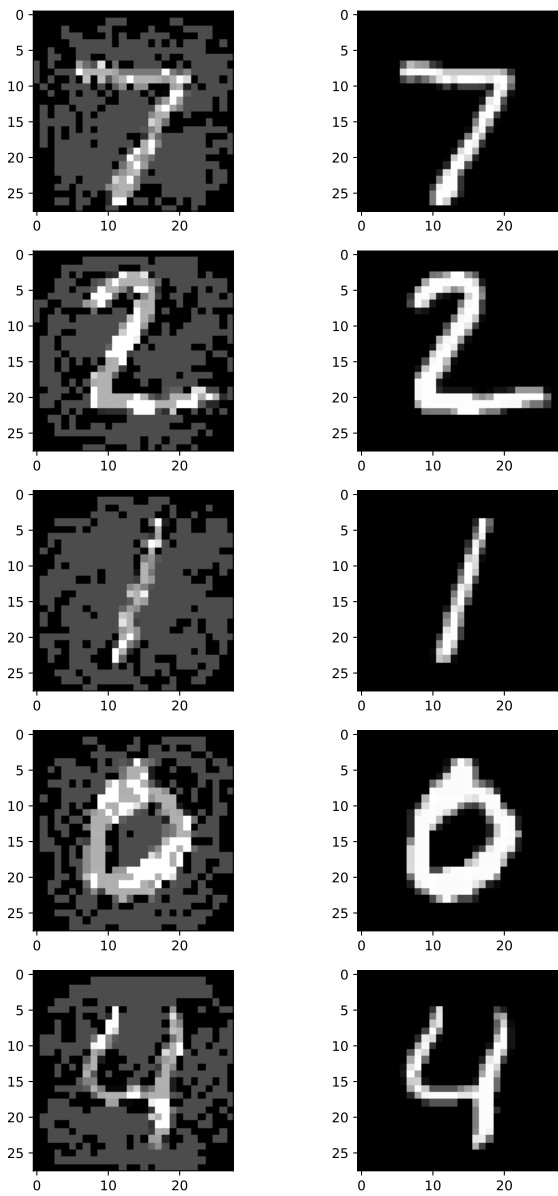

Figure 4: MNIST images. Perturbed images (left) and clean counterparts (right) from multiclass analytical kernel. $\epsilon = 0.3$

