# OpenReview forum: "The NTK Adversary: An Approach to Adversarial Attacks without any Model Access"
_ICLR.cc/2022/Conference — ICLR 2022 Submitted_

### Official Review · Reviewer_ttzq · 2021-10-30

**Correctness:** 1
**Technical Novelty And Significance:** 2
**Empirical Novelty And Significance:** 2
**Recommendation:** 3
**Confidence:** 4

**Main Review:**

My main concern with this paper is that its threat model is not clearly formulated, and in fact, is confusing. For example, when the paper says "Recall that PGD is a strictly more knowledgeable attack with access to the model parameters." this is not correct -- the threat model in this paper is in fact incomparable with the usual threat model in white-box attacks: The reason is that, for a big part of this paper, we assume that the adversary can access the training data (but not model parameters), but PGD attacks can access model parameters (but not training data).

Accessing training data is actually arguably even more knowledgeable -- as an example to this end, the whole field of differential privacy for ML is to study how to release a model, but where one clearly cannot access training data (then no guarantee whatsoever is there). Also, if an adversary can access training data, he can always retrain the model using that training data, and then PGD attack it. So I think allowing accessing training data is really strong.

Now this has left the choice of "training agnostic" method in this paper. However, now to this end when I look at empirical results in Table 4 on test data, then PGD (ce) is much better than NTK attacks. So I don't think I will buy the story that NTK attacks are really strong.

**Summary Of The Paper:**

This paper investigates to use NTK as a proxy to generate adversarial examples for neural nets. I like the paper in general. The derivations are clear and the experiments are clearly presented. However, I think this paper suffers significant problems from unclear formulation of the threat model.

**Summary Of The Review:**

A clear formulation and correct analysis of the threat model is one of the most important things in this kind of work,
and this work is somehow very confusing in this regard. Even though it has other merits, I'd recommend reject.

---

> ### Author Response · Authors · 2021-11-12
> **Reply to Reviewer ttzq**
>
> Thank you very much for raising this very interesting point. We agree that in the traditional "threat-guard" model it may seem that an  adversary with access to training data is more powerful than one that only queries the model. However, to respond to your points:
>  > "Also, if an adversary can access training data, he can always retrain the model using that training data, and then PGD attack it"
>
> The situation is a bit more subtle than that. Even if we train a different instantiation of the same model using the data and calculate  the gradients from it, the attack will be far less efficient compared to running PGD on the original model (see, for example Madry et al., https://arxiv.org/pdf/1706.06083.pdf Appendix B, Tables in Figure 8). So one cannot as straightforwardly claim that training a substitution model can be as powerful as having white-box access, as your comment might suggests.
>
> Even more crucially, we would like to argue that the generation of adversarial examples via PGD (or any white-box "attack") also *uses the training data*. Current research on adversarial machine learning focuses on a security paradigm, where a threat model is assumed, an attack is proposed, and subsequently a defense is created to mitigate the problem. In order to try to understand the origin of the existence of adversarial examples, it may be useful to abandon the distinction between an attacker and a defender and view their generation as a one way process. Viewed through this lens, for a PGD "attack", we train a model, the models learns from the data, and then by calculating the gradients of the model, we craft adversarial examples. Notice that this process uses all the information about the data that the model has stored in its weights: it **does** use the data. One contribution of our work is the observation that the weights of the model do not contribute to the generation of these examples (at least for the experiments conducted and those models in the NTK regime). Although this is already known to some extent from prior work that trains substitute models, we demonstrate that this notion holds from first principles that only require the a-priori description of the model.
>
> So, adopting the ML security paradigm indeed makes the two threat models incomparable, as you correctly point out, and our work might be confusing. However, from the viewpoint of trying to understand the generation of adversarial examples, a "PGD attack" is a **strictly** more knowledgeable procedure than the one introduced in our work. In some sense, the word "attack" may not be the most proper way to describe the methods introduced in this paper and the phenomena that we present. Instead, "NTK-attacks" may be usefully interpreted as an a-priori weakness of any model that fits into the stated assumptions.
>
> Your review helped us realize that parts of the paper can be emphasized to reflect our viewpoint with more clarity. After the feedback we received from your and the others reviewers' comments, we plan on working on this aspect on our revision, which we will share shortly.
>
> We would be grateful for your thoughts after this clarification of some aspects of our work.

---

### Official Review · Reviewer_jJ86 · 2021-11-01

**Correctness:** 3
**Technical Novelty And Significance:** 2
**Empirical Novelty And Significance:** 2
**Recommendation:** 5
**Confidence:** 5

**Main Review:**

While the proposed method is interesting and of practical importance, I have several concerns on performance evaluation on technical novelty compared to previous works, which are detailed as follows.

1. The authors mentioned in Sec. 2 that "To our knowledge, no prior work to ours has yet connected these
two areas to generate adversarial examples, or has exploited the parameter-free description the NTK
affords to devise attacks that do not require any information on trained model parameters."   I don't think this claim is true. In fact, the proposal of using NTK to construct adversarial attacks was already proposed [R1], an ICML 2021 publication. Thought the focus of that paper is on poisoning attack, [R1] also considers data perturbation, and the analysis looks quite similar to the proposed method. Since the perturbation analysis and the use of NTK is largely similar, I suggest the authors provide detailed comparisons to this work and discuss the novelty compared to [R1]. I also think the adversarial perturbation method used in [R1] can be a baseline method to be added in the performance evaluation.

2. Transfer attack without model access is known as "no-box attack" and there are several related works as well, but unfortunately the authors did not cite or compare to any of them.
The latest no-box attack paper that I know of is the NeurIPS 2020 publication [R2]. I suggest the authors compare to state-of-the-art no-box attacks on the studied settings so that one can better understand the effectiveness of NTK attack.


[R1] https://proceedings.mlr.press/v139/yuan21b.html
[R2] https://arxiv.org/abs/2012.02525


**Summary Of The Paper:**

This paper proposes the use of Neural Tangent Kernel (NTK) to generate transferrable adversarial examples without access to the target model.

The main contributions are:

1. The derivation of adversarial perturbation under NTK setting.

2. Illustration of different attack scenarios using NTK-attack (e.g. no training needed but several initializations, training-data-agnostic, etc).

3. Good transferability performance on MNIST and CIFAR-10 on the tested model compared to PGD attack (Table 4).

====== Post-rebuttal feedback

I thank the reviewer for providing the responses. However, I don't think they address my major concerns.

Technically, as the reviewer admitted, there is a similar paper published on the same topic (Yuan & Wu, 2021). Although (Yuan & Wu, 2021) focused on poisoning attacks, the fact that poisoning attack is essentially a bilevel optimization means it actual includes the studied inference-attack on NTK as a special case, by treating the target model as fixed and setting the time to infinity in NTK (as the authors also pointed out). In addition, I don't think doing a first-order approximation then solving the approximated problem is a "difference" as the authors' tried to argue; to me, this will lead to a suboptimal solution, not even an innovation.

Contribution-wise; applying adversarial attack on NTK at inference time seems a trivial extension of PGD attack, as other reviewers also mentioned. The considered transfer attack does not have a solid demonstration of why the result is significant and does not compare to SOTA transfer attacks in the same setting, like no-box attack.

**Summary Of The Review:**

This paper's idea is interesting, but the main concerns are

1. Technical novelty compared to existing works on existing NTK-based generalization attacks. The authors did not discuss the differences at all.
2. Lacking performance comparison to state-of-the-art no-box attacks.

---

> ### Author Response · Authors · 2021-11-12
> **Reply to Reviewer jJ86**
>
> Thank you very much for critically raising concerns about the novelty of our work. We address your two points:
>
> 1. You are right, in light of the reference [R1] you provide, this claim is incorrect, together with some others made in the introduction of the paper. We were not aware of this work, whose publication, in our defense, was almost concurrent to the preparation and submission of our own paper. We are glad that aspects of this line of research are of interest to the research community and have been explored, albeit from a different point of view. We will correct our statements in the revised version, but let us now comment on the essence of your point, namely the differences between our work and [R1]:
>
>     1) [R1] uses similar tools (the NTK as a proxy), but solves a different task and draws very different conclusions. That work is concerned with what is coined as *generalization attacks*: the process of altering the data distribution so that a model being trained on it cannot generalise on clean data. In contrast, our work analyses *trained* models to analytically derive their weakness to adversarial perturbations. [R1] finds perturbed data to *train* models by using what we call the empirical kernel. In other words they solve the following problem (following the notation of our work, c.f. Sec. 3.1):
>     \begin{equation}
>         \max_{\epsilon} \mathcal{L} \left(\mathbf{H}(\mathcal{X}, \mathcal{X} + \epsilon) \mathbf{H}(\mathcal{X} + \epsilon, \mathcal{X} + \epsilon)^{-1} (\mathbf{I} - e^{-\lambda \mathbf{H}(\mathcal{X} + \epsilon, \mathcal{X} + \epsilon)t}) \mathcal{Y}, \mathcal{Y} \right),
>     \end{equation}
>     where $\lambda$ is the learning rate of the training procedure and $\mathcal{L}$ a loss function. This method is about the *training* phase of a model. Our work, which is instead concerned with the *inference* phase, solves the following problem (we only consider the case of converged models, i.e. $t \to \infty$):
>     \begin{equation}
>         \max_{\epsilon} \mathcal{L} \left(\mathbf{H}(\mathcal{X}^\prime + \epsilon, \mathcal{X}) \mathbf{H}(\mathcal{X}, \mathcal{X})^{-1} \mathcal{Y}, \mathcal{Y}\right).
>     \end{equation}
>     In other words, we assume the existence of a trained model and then generate adversarial examples that maximally fool it. Note that in our work, we cover both the cases where $\mathcal{X}^\prime = \mathcal{X}$ and, the more general one $\mathcal{X}^\prime \neq \mathcal{X}$ (see Section 3.1, Subsection ``Adversarial perturbations of unseen (test) data"). Therefore, although both methods use similar tools, they solve different problems.
>
>     2) [R1] solves the equation above via the "standard" route: projected gradient ascent on the loss, which is a first order approximation of the optimization problem. Instead, in our analysis, we leverage the closed form expressions that the NTK theory provides to propose a different approach: first take a first-order approximation of the *function* of the model, and then solve the maximization problem directly (without linearizing the loss). We have already commented on this subtle point in the last sentence of Section 3 in our paper. The fact that this "swapped" solution of the underlying optimization problem produces high-quality adversarial examples is a significant part of the novel contributions of our work, that is completely absent from [R1], and can not be established in general without the connection to the NTK that we introduce. Moreover, this analysis allows to obtain expressions for the perturbed data and the output of the network on them, something that does not follow from the methods of [R1] (even if extended to the problem we consider), because it only considers empirical versions of the kernels.
>
>  Once again, we want to thank you for raising this concern, because it gives us the chance to clarify potential confusions in our soon-to-be-uploaded revised version and to carefully emphasize the many novel parts of our work.
>
> 2. We will add references to the "no-box" setting, which is indeed very relevant to our discussion. It could be valuable to assess our method in very small data regimes to match the assumptions of the "no-box" setting that [R2] considers and then compare the results. However, please note that [R2] trains substitute models in order to generate adversarial examples, which is in contrast with our method which does not require any training (with usual gradient descent techniques) of substitute models. This was the reason why we evaluated our attacks against "PGD attacks", and did not proceed with an extensive comparison with the large body of work that exists on "black-box attacks".
>
> We hope the above argumentation, when included in the revised paper, covers your two concerns.

---

### Official Review · Reviewer_BKjF · 2021-11-01

**Correctness:** 3
**Technical Novelty And Significance:** 3
**Empirical Novelty And Significance:** 2
**Recommendation:** 5
**Confidence:** 4

**Details Of Ethics Concerns:**

Adversarial Attacks are known problems for Deep Learning models. They can be used to cause significant harm in a range of situations, such as autonomous driving or health care. Research into making these attacks cheaper and/or more scalable is hence a risk as it lowers the barrier for potential misuse.


**Main Review:**

The paper is clearly written and the contributions well put into context of existing research. The motivation as well as the line of argumentation are well chosen. The authors first describe the challenges with current attack-models in the white- and black-box setting and motivate why NTK based adversarial attacks are able to circumvent these challenges. They do briefly address the shortcomings of NTK-based attacks, needing access to the training data and labels and defer detailed studies on how to address these shortcomings with test data and pseudo-labels  to subsequent work. The authors give theoretical background on how to think about the NTK attacks and how to choose the ideal NTK attack in the binary- and multi-class classification setting based on a first order Taylor expansion of the closed-form NTK. The math appears sound and well presented with more details presented in the appendix. To validate the theoretical argumentation the authors provide experimental evidence on MNIST and CIFAR-10 for a variety of models (mentioned above) that fall within the limit of NTK and far away from the NTK regime and show that at least anecdotally these attacks can work and transfer to other models.

While the line of argumentation is well chosen, this reviewer has a few open questions about the solidity of the argumentation.
- To begin with it is unclear why a first order Taylor expansion is sufficient for the NTK attacks being considered here. It would be good to understand the theoretical assumptions that need to be made in order to show that the first order correction is always finite and non-vanishing  and at the same time dominating the second order corrections.
- While the sufficiency of the first order correction is shown before based on proofs of Chizat et al. https://arxiv.org/abs/1812.07956, these proofs only apply to the weights of the networks not the inputs. As a consequence the generalization to first order only input corrections are not obvious.
- Moving on to the experimental validation, the reviewer wonders about the validity of the empirical NTK underlying the data in Table 1. Looking at the vast performance differences between the analytical and empirical NTK it would be good to see a trendline for the empirical kernel as a function of batch-size to see that the empirical kernels indeed do converge to the analytical one. This is important (i) as a sanity check for the experimental procedure and (ii) an understanding of how computationally costly the empirical NTKs are for a given attack success-rate. Especially point (ii) is important when comparing to black-box query attacks that have high sample complexity to approximate the empirical gradient.
- Looking at Figure 1 the differences between NTK-based and PGD-based attacks are stark. While it is to be expected that the PGD-attack will always be dominant, this reviewer wonders about the different contributions to the gap. To be more specific: what are the effects of the imperfect  empirical NTK due to finite sampling size  and how much does neglecting the second-order contributions to the Taylor expansion contribute? Could this gap be closed by incorporating either of the two? As an intuition the  gap should certainly be close to zero if the empirical kernel is close to the analytic one and we consider 1-step PGD.
- Looking at Table 3, this reviewer is concerned about some uncommented trendlines: Looking at depth=2 the NTK success-rate diminishes with increasing width. At depth=4 the performance shows a goldi-lock behavior as a function of width, while the trend at depth=6 is the opposite of depth=2. It seems like there are some unexplained patterns and the authors should address this through either theory or more experimental ablations.

Finally some minor points for improvements:
- The authors state at several occasions that they study the effects over several random seeds. If so, they should also report and plot uncertainties in all places or clarify which experiments have been run with random seeds. So far this is only done in the binary-MNIST case of Table 1.
- It would be helpful to the reader if the unperturbed baselines would be reported in Table 4 as well to put the attack strength into context.
- While the authors clearly defer the study of efficacy of test-data with pseudo-labels for NTK-based attacks, it would be good to state why the authors believe this approach to work. As of now there is no reason to believe the attack will work in this setting, putting serious restrictions on the efficacy of the NTK-based attack.
- Finally, a different way to view these results is through the lens of verifying/falsifying NTK theory. The approach to NTK theory is novel and intriguing and can lead to interesting new findings. In this light, however, this reviewer would like to challenge the notion that the findings are so unexpected: IF NTK theory is valid then many networks learn a similar kernel in the limit. Finite-size corrections should be small and hence NTK-based adversarial samples should be able to fool a range of models that are a finite-size correction away from the NTK limit. On the other hand, if the findings do contradict NTK theory, then this will lead to intriguing new paths of research to understand the breakdown of this theory.


**Summary Of The Paper:**

In this paper the authors introduce and study a new approach to generate Adversarial Attacks for image classification models. The core of their approach lies in leveraging Neural Tangent Kernel (NTK) formulations of Deep Neural Networks. By using the analytic and empirical NTK versions of common network architectures they are able to generate adversarial samples without requiring white- or black-box access to the model under attack. The work establishes a new link between formerly disjoint research fields of Adversarial Attacks and NTKs.
They show empirically that attacks generated using a simple model with closed-form, analytic NTK transfer reasonably well to other models that are far from the kernel regime. In addition they also study the adversarial samples generated from an empirical, sampled kernel via auto-differentiation and show their success in attacking other models.
The authors provide ablations on MNIST and CIFAR-10 for standard models such as the analytically solvable Random Combination of Features and deep models such as Fully Connected Networks and Convolutional Networks in the LeNet-family.

**Summary Of The Review:**

In summary, the paper introduces a new type of adversarial attacks, based on NTKs and exploiting the transferability of adversarial samples between models. The authors clearly motivate the relevance of their work and chose a suitable line of argumentation consisting of theoretical as well as empirical evidence. However, this reviewer is missing some rigor in the theoretical underpinning as well as the experimental ablation. The work leaves open a few key questions about the validity of arguments, specifically the connection between the NTK limit and the models far away from the NTK limit,  and as of this point shows anecdotal evidence of the method being generally applicable. This is heightened by the fact that the authors are looking at very special datasets as well, MNIST being small and relatively easy and CIFAR-10 being small and complex. It is unclear how these attacks would fare on larger, production-scale models, trained on, e.g., Imagenet. Specific points to improve upon are:
1. Solidify the theoretical underpinning by providing intuitions and argumentations, possibly proofs, why the second order contributions are negligible
2. Show that their experimental setup indeed can reproduce the analytical kernel by doing ablations with respect to the sample size and show that the kernel tracks towards the limit and at what cost.
3. Addressing points 1 and 2 can then give some understanding of the contributions to the gap in performance between PGD and the empirical NTK-based attack curves
4. Provide some understanding / insights about the different scaling directions of table 3, i.e. why do the directions reverse order as depth increases
5. Give some intuitions / early evidence why this work would translate to test-data with pseudo-labels, as this is the most likely and relevant scenario in practice.

Based on these observations this reviewer cannot recommend the publication of this work, despite its very intriguing line of research.

---

> ### Author Response · Authors · 2021-11-12
> **Reply to Reviewer BKjF (1/2)**
>
> Thank you so much for your careful assessment of our work and your detailed feedback. We agree with several points raised in your review, but let us clarify our position on them.
>
> 1)
> > "To begin with it is unclear why a first order Taylor expansion is sufficient for the NTK attacks being considered here [...] are not obvious."
>
>     This is a very constructive point and a non-trivial detail of our argumentation. The rationale behind the first order Taylor expansion is, however, the same that applies to the universally adopted approach to finding adversarial examples, i.e. projected gradient descent: $2$-nd order terms have a $O(\eta^2)$ factor, which, given the fact that we are interested in small perturbations $\eta$, can be neglected. Of course, there is the subtle point (on which we already commented in the last sentence of Section 3 of our paper) that we first linearize the function of the model, and then solve the maximization problem exactly, while in PGD one maximizes the linearized optimization problem. While we agree that a more rigorous analysis will benefit the understanding of the problem, we feel that our contributions rely on similar assumptions as adopted by the community in prior work. We will expand on this argumentation in our revised paper, and we will consider adding more theoretical arguments.
>
> 2) >"Moving on to the experimental validation [...] for a given attack success-rate."
>
>     Thank you for suggesting this sanity check. First, allow us to be precise and clarify that the performance difference between the analytical and empirical NTKs in Table 1 is vast only in the case of CIFAR, but not for MNIST. With our current implementation, it is computationally expensive to estimate the empirical kernels with batch sizes larger than 1500, which is what is currently reported. Please see our detailed reply to reviewer **CStK** for the discussion on scalability; in a nutshell, we expect that this aspect of our work  can be expedited moving forward if other frameworks are adopted. For the time being, one immediate experiment we plan to implement to incorporate your helpful suggestion in our revised version is to compute the analytical kernel using smaller batch sizes, to get a trendline that at least covers point (i).
>
> 3) >"Looking at Figure 1 the differences between NTK-based and PGD-based attacks are stark. [...] Could this gap be closed by incorporating either of the two?"
>
>     This point is similar in spirit to the previous one. Please allows us, again, to be precise and remark that the differences are large in only one of the two cases, i.e. on MNIST this time. We agree that it would be insightful to disentangle the effects that different assumptions have on the accuracy of the attack. For that reason, we will reproduce Figure 1 (b) for the analytical kernel. Thank you for this useful suggestion.
>
>  4) > "Looking at Table 3, this reviewer is concerned about some uncommented trendlines: [...] theory or more experimental ablations."
>
>     We agree there are some interesting patterns in this experiment, but we respectfully disagree that those are uncommented. We state (in an admittedly brief fashion, given space constraints): "We observe for deeper models (depth = 4 or 6) that with increasing width the gap to the random baseline widens". By that we mean to argue that looking solely at the NTK success rate might be misleading, since the robustness of the models on random noise also changes (loosely speaking, we know that the capacity of the network affects its robustness). So, if instead we focus on the ratio between the NTK and the random baseline, then the numbers are:
> | depth      |  width (100, 1000, 10000) |
> | ----------- | ----------- |
> | 2      |    (0.31, 0.34, 0.36)     |
> | 4   |    (0.39, 0.34, 0.31)        |
> | 6  |  (0.63, 0.47, 0.37)         |
>
> For depth equal to 4 and 6, we see, clearly, a better relative success rate of the NTK attack as the width increases, which matches our expectations since the model is expected to operate closer to its tangent kernel with larger width. We currently do not have a good explanation on why this is not observed for depth equal to 2, although one should not read too much into this experiment from a theoretical perspective, since, as stated, these models were trained violating many of the NTK assumptions. We believe the above sufficiently addresses the results of the experiment. We kindly ask you to provide feedback on this explanation, so that we can take it into account in the revised version and perhaps extend our discussion of this experiment.

---

> ### Author Response · Authors · 2021-11-12
> **Reply to Reviewer BKjF (2/2)**
>
> Thank you for bringing up the minor points. We will consider making the related micro adjustments in our revised version. On the two most important of them:
>
> > "While the authors clearly defer [...] work."
>
> The experiments conducted demonstrate the efficacy of the NTK attacks on test data (both with and without access to the training data). The case of having no access to test labels is covered in our subsection ``Absence of labels", and we argue that, in light of our experiments, it will also be successful. This is based solely on the fact that an accurate target model will provide  the correct labels with high probability, and given that this will be the only change in equation (4), it is reasonable to assume that the attacks will also be effective with high probability. We firmly believe that such experimentation, as interesting as it might be, is not important for establishing the main contributions of our work and it may create confusion, since it shifts the threat model (it assumes oracular access). This was the reason why we deferred its study.
>
> >"Finally, a different way [...] this theory.":
>
> Thank you for recognizing the novelty of our approach and its potential in providing new insights, and bringing up this excellent point. Indeed, we agree that in hindsight the findings might not be so unexpected, especially the ones on models that operate in the NTK regime. These observations, however, have non-trivial implications for what we know about adversarial examples: to some extent, our work suggests that even before the training procedure starts, we know, *a-priori*, what the adversarial weaknesses of the model will be, given that our attacks use only information that is available at initialization - no information about the weights of the model whatsoever. Although this is already hinted at in prior work that trains substitute models and computes adversarial data from them, our work demonstrates that this notion holds in a more model-specific way than what was assumed before, and its connection to kernel methods may shed some light to the puzzling transferability phenomenon. Trying to disentangle the different factors and argue about the validity of the NTK theory under this lens is an excellent future direction.
>
> Since the points you made were very concrete, we kindly ask you to provide feedback on whether our clarifications and arguments were sufficient and clear enough, so that our revision includes them.
>
> Last but not least, we would like to comment on the ethical concerns you raised. We believe that adversarial examples constitute a real problem for machine learning deployment, both from a security perspective and the perspective of interpretable models that match human perception. As such, researching this problem, providing insights and submitting the results for publication, thus making them available to the scientific community, is the only ethical choice for researchers. Not researching or not publishing the findings leaves space for actual adversaries to be the first who acquire the knowledge and use it to exploit these weaknesses. We hope our arguments will make you consider lowering the ethical flag. Thank you.

---

### Official Review · Reviewer_CStK · 2021-11-03

**Correctness:** 3
**Technical Novelty And Significance:** 2
**Empirical Novelty And Significance:** 2
**Recommendation:** 3
**Confidence:** 3

**Main Review:**

The major concerns are as follows,

(1) I am afraid that It is difficult for the proposed method to scale to larger models and datasets, which restricts its application in practical black-box attacks.

-(1.a) On a real-world dataset, the number of training data is extremely large (e.g., millions for ImageNet-1k). However, I conjecture that the proposed requires the full training data to craft adversarial perturbations since the absence of full data on CIFAR-10 is devastating as discussed in Section 4.1.1.

-(1.b) The method seems to be extremely slow. Even if crafting a single adversarial example, we have to calculate gradients for every training sample with respect to random parameters, and then average the results over differently sampled parameters according to Eqn. (3). The number of training samples is large. Besides, I guess the number of sampling might be larger due to the low sampling efficiency in extremely large parameter space (e.g., ResNet-50 on ImageNet). This is almost prohibited.

-(1.c) The CIFAR-10 empirical results are only based on toy models, much smaller than commonly-used models like VGG-19 or ResNet-18, which can't convince me.

(2) Since this paper never proposes a practical attack method nor helps me understand adversarial attacks deeper, I am afraid this paper lacks contributions to the field of adversarial examples.

I am a researcher who mainly focuses on adversarial examples and is unfamiliar with NTK. If any reviewer or the authors could convince me that this paper contribution enough to NTK, I would be glad to reconsider my rating.

**Summary Of The Paper:**

This paper proposes an adversarial attack that does not require any access to the model under attack or any trained replica of it with the help of NTK. The authors only demonstrate the effectiveness of the proposed method on small data and toy models.

**Summary Of The Review:**

Although this paper explores an interesting problem, the proposed method is almost infeasible and the contributions are really limited. As a result, I don't think this paper achieves the requirements of ICLR.

---

> ### Author Response · Authors · 2021-11-12
> **Reply to Reviewer CStK (1/2)**
>
> Thank you very much for your concerns on the scalability of the proposed methods, and whether our findings will hold on larger models. In addition to the remarks already  made in the paper about computational issues, we address your points:
>
>  > "[...] restricts its application in practical black-box attacks".
>
> Please note that the setting of this work is different than the one assumed in `"black-box attacks". Our paper shows that it is possible to generate adversarial examples, *without* querying the actual, trained, model and *without* training a substitute neural network. This is, of course, the reason why we do not perform an empirical comparison of the proposed methods with the large body of work that exists on "black-box attacks" and only compare with the best known method of producing adversarial examples, which is projected gradient descent. Therefore, we argue that scalability is not the most pertinent aspect of our work, since its focus is not on introducing a practical "black-box attack", but rather using theoretical arguments to present an interesting phenomenon that is valid under a setting different to the "black-box" setting. With that being said, while we share some of your concerns on the computational expense of creating adversarial examples with our methods,  allow us rebut a few of your arguments.
>
> 1) > "I conjecture [...] Section 4.1.1"
>
>     For the time being, this is a possible but far from verified conjecture. Note that Section 4.1.1 presents experiments on a simple model, coming with a simple kernel, deliberately chosen this way as a first proof-of-principle. Our methods rely on the representative power of the kernels in order to generate the perturbations. It is possible that more complex kernels such the ones that stem from convolutional architectures (CNTKs) may be able to capture data dependencies using few samples (see for example https://arxiv.org/pdf/1910.01663.pdf for a performance evaluation of such kernels in low-data regimes). In addition, please note that our experiments on MNIST demonstrate that small batches (few data) are enough to create high-quality adversarial examples. One may argue that MNIST is an easy dataset to learn, but it also holds that it is more difficult to attack with small perturbations given the better separability of its classes. So, in general, we do not believe your statement holds, but would be curious to hear your arguments on why it could be true, given the arguments presented.
>
> 2) > "Even if crafting a single adversarial [...] This is almost prohibited."
>
>    Please note that this is far from true in the case of analytic kernels, which are computable for a wide variety of models. Their expression allows very fast generation of adversarial examples (a matter of seconds or minutes for MNIST and CIFAR), since it only involves a few tensor multiplications.
>
> A general comment regarding your concerns about scalability: the experiments with the empirical kernels were run with non-optimized auto-differentiation tools on PyTorch, and the focus was not on scalability, but on establishing proofs-of-principle, since the setting is different than the one usually assumed by an adversary. Thanks to a reference that reviewer **jJ86** pointed out to us, we add that the code developed in [R2], based on JAX and the Neural Tangent library, if adjusted to the problem we consider in this work (see our reply to **jJ86**), can also extend generation of adversarial examples to larger models (ResNet) and datasets (ImageNet). We agree that performing extensive experiments to check under what conditions our findings still hold and how much the theoretical assumptions on width, depth, scale of initialization, activation function, preprocessing of the data etc. matter is an important venue for future work, which, however, places us into the intimately connected ongoing debate on the validity of NTK theory, which almost three years after its introduction is still open. Finally, the computational aspects of computing NTK related quantities is still an active area of research (see https://openreview.net/forum?id=zLb9oSWy933 for an ICLR 2022 submission), and we expect advances in that area to enable further experimentation with the methods that we introduced in this work.

---

> ### Author Response · Authors · 2021-11-12
> **Reply to Reviewer CStK (2/2)**
>
> More importantly,
>
> > "Since this paper never proposes a practical attack method nor helps me understand adversarial attacks deeper ..."
>
> As already stated, the main focus of this work is not to devise a practical black-box attack, as its setting is different than the so-called black-box setting. We would like to argue that one of our main contributions lies in the latter part, namely an enhanced understanding of adversarial attacks. Our experiments suggest that for the models that lie within the NTK regime, the weights of the model contribute nothing to the generation of adversarial examples. To some extent, our findings suggest that even before the training procedure starts, we know, **a-priori**, what the adversarial weaknesses of the model will be. Although this is already hinted at to some extent in prior work that trains substitute models and computes adversarial data from them, our work demonstrates that this notion holds in a more model-specific way than what was assumed before, and its connection to kernel methods may shed some light to the puzzling transferability phenomenon. See also our reply to reviewer **ttzq**, for further discussion of this aspect. We want to thank you for this specific comment, because it made us realize that some statements inside the paper may not sufficiently emphasize this important contribution of our work. Our revised version will be more explicit in that regard.
>
> With respect to the NTK-related contributions of the paper, we believe that the connection between NTK and generation of adversarial examples at inference time together with its theoretical analysis is valuable and provides several new ways to argue about adversarial examples, an opinion that reviewer **BKjF** also seems to share. Additionally, please consider reading Point 1.1 of our reply to reviewer **jJ86**, since their review brought this specific aspect of our work into discussion, and kindly comment again if you need further clarifications.

---

### Official Review · Reviewer_YPLb · 2021-11-11

**Correctness:** 3
**Technical Novelty And Significance:** 2
**Empirical Novelty And Significance:** 2
**Recommendation:** 3
**Confidence:** 4

**Main Review:**

This paper derived a few ways to attack the deep neural networks in the kernel (NTK) regime. This paper derived a few ways to attack the deep neural networks in the kernel (NTK) regime. The authors performed experiments on single-layer models and certain specific deep models on MNIST and CIFAR10 using the proposed attacks.

The main novelty of this paper is the proposed NTK-based attack, which attempts to connect the theory (what we understand about the large width limit) and practice (whether a deep neural network is robust). However, several weaknesses prevented me from recommending acceptance, detailed below.

(1) Technical novelty: while the reviewer wasn't aware of any NTK-based attacks, the proposed attack is a very straightforward extension of the well-known results (essentially, just simply linearizing the model using the kernel and applying FGSM).

(2) Empirical usefulness: from my reading, I don't think the proposed attack is widely applicable and has a very limited impact:

(2a) The proposed attacks are derived from the (large width, small learning rate) limit of deep networks under *standard* training, i.e. no adversarial training. It is well-known that adversarial training is nearly necessary for adversarial robustness - while both the perturbation derived in section 3 and the models evaluated against in section 4 are not adversarially trained. It may be interesting to see whether the analysis in Gao et al. 2018 about the limit of adversarial training in the large width regime can shed light on the attack of adv-trained models.

(2b) The proposed attacks do not produce a way to better defend against adversaries. It is well-known that attacking DNNs is much easier than defending, so finding a new attack is much less interesting than finding a new defense.

(2c) The experiments in section 4 are evaluated against "toy" models that are very different from near-SOTA models like TRADES and its variants. As mentioned in 2a, none of these models are adversarially trained.

(2d) Other technical limitations: e.g. all of the models in section 4 are binary-classification models; some of the results require the adversary to have access to the training data/labels, etc.

**Summary Of The Paper:**

This paper derived a few ways to attack the deep neural networks in the kernel (NTK) regime. The authors performed experiments on single-layer models and certain specific deep models on MNIST and CIFAR10 using the proposed attacks.

**Summary Of The Review:**

I vote for rejection since I don't think the technical novelty and empirical usefulness of the results have reached the level of ICLR paper.

---

> ### Author Response · Authors · 2021-11-13
> **Reply to Reviewer YPLb**
>
> Unfortunately, we fear that this late review might reflect a precipitated reading of our work, since it seems to miss the entire setting of our paper, crucial details from the methods and the experiments, and possibly misreads its conclusions. Let us elaborate:
>
> 1) > "the proposed attack is a very straightforward extension of the well-known results (essentially, just simply linearizing the model using the kernel and applying FGSM)"
>
>      We are surprised by this sentence, since it seems to be obviously incorrect.
>
>      There are several misconceptions present in this statement, but, briefly, FGSM is a method that runs one step of projected gradient ascent (equivalent to maximizing a first order approximation) on the loss of a neural network, while our work solves the same optimization problem on a converged infinite neural network by taking a first order approximation on the model's output *function*, and then solving the maximization problem exactly. We agree that the math looks similar at first glance, though.
>
> 2) > "The proposed attacks are derived from the (large width, small learning rate) limit of deep networks under standard training, i.e. no adversarial training [...] none of these models are adversarially trained."
>
>     Points (2a, 2c) argue that the proposed methods are not interesting or evaluated incorrectly, given that they are run against models that are not adversarially trained. To the best of our knowledge, "black-box attacks" in most cases demonstrate their effectiveness on undefended models. So, the subset of our work that applies to this threat model follows the same rules. In our paper, we only analyze non-adversarially trained models, because we present the implications of NTK theory to the non-robustness of models. We clearly state this in the introduction of our method (the kernel description applies when we train with gradient descent (flow) on the $\ell_2$ loss), in the experiments, and, finally, in the discussion, where we explicitly say why this is an interesting setting. After constructive comments from the other reviewers, we will expand on this discussion in our revised version of the paper.
>
> 3) > "all of the models in section 4 are binary-classification models"
>
>     This is wrong. Sections 4.1.2, 4.2 and 4.3 very explicitly contain experiments on multiclass classification tasks.
>
>
>
> Apart from possible sloppiness in the evaluation of the details of our work, we feel this reviewer misses the point. We devise an attack that operates at inference time based on  NTK theory and uses no information about the weights of the model. Our work uses theoretical arguments to reveal an interesting phenomenon: trained neural networks have a weakness, in the form of susceptibility to adversarial examples, that does not depend on their specific weights, but instead these blind spots can be calculated **a-priori**, at initialization, even before the training starts. For more, we would respectfully ask to please **carefully** read our work and the other reviews.

---

### Public Comment · ~Chia-Hung_Yuan1 · 2021-11-10
**Novelty and Missing Citation**

In the last paragraph of Section 2, you mentioned that "However, to our knowledge, no prior work to ours has yet connected these two areas to generate adversarial examples, or has exploited the parameter-free description the NTK affords to devise attacks that do not require any information on trained model parameters."

Unfortunately, this paper does not cite adequate related works, particularly regarding the interaction between NTK and the attack, which has been proposed in Yuan, Chia-Hung, and Shan-Hung Wu. "Neural Tangent Generalization Attacks." International Conference on Machine Learning. PMLR, 2021.

---

> ### Author Response · Authors · 2021-11-12
> **Reply to "Novelty and Missing Citation"**
>
> Thank you very much for your comment and on your contribution to the public discussion of this paper. Indeed, we were not aware of your work, and we will add references to the revised version of the paper. Reviewer **jJ86** already mentioned this related work, and in our reply to their review, we address briefly all the points where our work differentiates from it. Please consider reading our reply there and comment again, if you need any clarifications. Extensive discussion will be added to the revised paper.

---

### Author Response · Authors · 2021-11-22
**Summary of changes in the revised version**

We would like to thank all the reviewers for the time spent reading and understanding our paper, and for their constructive comments that helped us improve our work in the revised version.  In summary, wetried to clarify the scope of the paper, and included significantly improved results using analytical kernels. We outline the main changes of the new version:
1.  We improved the presentation of the main contributions of our work, and in particular emphasized the implications of our experiments on what we know about adversarial examples in the overparameterized regime (see "Our contribution”, p. 1).
2.   We  included  results  for  the  multiclass  classification  experiments  with  the  **analytical**  kernel  of the RCF model, and demonstrate improved results that almost match the performance of PGD on the experiments presented (Sec.  4.1.2 and 4.3).  Previous results with empirical kernels now moved to the Appendix (A.3.3).  We also included error bars in all the random baselines.  We want to thank reviewer **BKjF** for suggesting these changes to improve our work.
3.  We added references to a closely related prior work, that we missed in our first version and we are grateful to reviewer **jJ86** for bringing it up.  We made minor changes in the main text, so that we are precise in our statements regarding the novelty of our approach in light of this new reference, and discuss the differences between the two works in the end of Sec.  2.  We also added references to the “no-box” setting (p. 3, before the Neural Tangent Kernel paragraph).
4.  After a suggestion of reviewer **BKjF**, we added ablation studies with batch size on the success ofthe attacks, that also serve as sanity check for the results presented with the empirical kernel.  They give a more complete picture of how different factors contribute to the success of an attack (Appendix A.3.4).
5.  We added Appendix A.3.5, addressing the main concerns of reviewer **CStK** on scalability.  Briefly, we mention explicitly that computational time is not an issue for analytical kernels, and we discuss the best  way  moving  forward  for  performing  experiments  with  empirical  kernels  of  large  scale  models  and datasets.
6.  We added Appendix A.4, addressing an interesting point that reviewer **ttzq** brought up regarding knowledge assumed by a white-box adversary vs an adversary in our work.  Briefly, we claim that, for the purposes of our paper, it holds that generating adversarial examples with projected gradient descent is strictly more knowledgeable than our method.

Again, we wish to thank all reviewers for their constructive comments, which allowed us to be more clear about the novelty, scope and implications of our work, as well as the new avenues for exploration it opens. Together with the significantly stronger results from analytical kernels in the multiclass setting, we truly hope that you will share our belief that our paper in its current state matches the standards of ICLR.

---

### Decision · Program_Chairs · 2022-01-20

**Decision:**

Reject

**Comment:**

The paper relies on the analytical tools afforded by on the NTK theory to proposes an adversarial attack that uses the information of the model structure and training data, without the need to access the model under attack. While the reviewers found the problem interesting and well motivated, they feel that the theoretical analysis and the experimental results can be significantly improved. In particular, some of the points that the reviewers did not find convincing during the discussion include: (1) the technical novelty of the work, i.e., applying adversarial attack on NTK at inference time seems a trivial extension of PGD attack; (2) authors' argument that knowing the model is strictly stronger than knowing the original training data; (3) scalability and generalization of the proposed method to settings without training and test set; and (4) comparison to existing sota transfer attacks in the same setting, like no-box attack. Addressing the above points will significantly improve the manuscript.